# Benchmarking strategies for cross-species integration of single-cell RNA sequencing data

Yuyao Song [1] ✉, Zhichao Miao [1,2], Alvis Brazma [1] & Irene Papatheodorou [1] ✉

The growing number of available single-cell gene expression datasets from different species creates opportunities to explore evolutionary relationships between cell types across species. Cross-species integration of single-cell RNA-sequencing data has been particularly informative in this context. However, in order to do so robustly it is essential to have rigorous benchmarking and appropriate guidelines to ensure that integration results truly reflect biology. Here, we benchmark 28 combinations of gene homology mapping methods and data integration algorithms in a variety of biological settings. We examine the capability of each strategy to perform species-mixing of known homologous cell types and to preserve biological heterogeneity using 9 established metrics. We also develop a new biology conservation metric to address the maintenance of cell type distinguishability. Overall, scANVI, scVI and SeuratV4 methods achieve a balance between species-mixing and biology conservation. For evolutionarily distant species, including in-paralogs is beneficial. SAMap outperforms when integrating whole-body atlases between species with challenging gene homology annotation. We provide our freely available cross-species integration and assessment pipeline to help analyse new data and develop new algorithms.

Animal cells show conspicuous diversity and have fundamental unities across species. Recently, single-cell RNA-sequencing (scRNA-seq) and single-nucleus RNA-sequencing (snRNA-seq) have been applied in a diversity of species, generating full-body cell atlases from transcriptome profiles of millions of individual cells[1–7]. Comparing cellular expression profiles cross-species provides insights into the origin and evolution of organs and cell types[6,8–10], highlights species-specific expression patterns[11,12], as well as further delineates how cell types execute their functions in health and disease[13,14].

Due to millions of years of evolution, the gene expression profiles of evolutionarily related cell types from different species exhibit significant global transcriptional shifts[11,15,16]. To compare these profiles, a cross-species mapping of genes via sequence homology is necessary to place cells into the same expression space. However, when performing a joint analysis of scRNA-seq data on the raw count data, a "batch effect" emerges between species. Cells from the same species tend to exhibit higher transcriptomic similarity among themselves rather than with their cross-species counterparts[16,17]. We term this "species effect", to distinguish from pure technical batch effects between same-species samples[18]. By correcting for the species effect during joint analysis of scRNA-seq data, we can find a latent representation of cells from different species on which they form clusters based on transcriptomic identity. Integrating scRNA-seq data cross-species can therefore identify cells that are transcriptomically similar between species, informing the degree of cell type expression conservation and divergence, as well as generating hypotheses about evolutionary cell type homology[10,19].

[1]European Molecular Biology Laboratory-European Bioinformatics Institute (EMBL-EBI), Wellcome Genome Campus, Hinxton CB10 1SA, United Kingdom. [2]Guangzhou Laboratory, Guangzhou International Bio Island, Guangzhou 510005, China. ✉e-mail: ysong@ebi.ac.uk; irenep@ebi.ac.uk

Various data integration algorithms originally designed for batch correction or cross-condition integration in scRNA-seq have been rapidly adapted to cross-species analysis[16]. A recent benchmark of batch correction methods[17] highlighted top-performing algorithms including: mutual nearest neighbours based fastMNN[20]; iterative clustering based Harmony[21]; LIGER which utilises integrative non-negative matrix factorization (iNMF)[22] and its recent upgrade LIGER UINMF that also takes unshared features[23]; a panorama-stitching algorithm Scanorama[24]; a probabilistic model with distributions specified by deep neural networks scVI[25] and its semi-supervised extension scANVI[26]; as well as SeuratV4 methods, which uses canonical correlation analysis (CCA) or reciprocal principal component analysis (RPCA) to identify anchors between datasets then uses dynamic time warping to align the subspaces[27–29].

Although a diversity of batch-correction algorithms is available, it is still very challenging to generate an informative cross-species joint embedding[16,17,19]. On one hand, species effects can be much stronger than average technical batch effects, so moderate methods frequently fail to integrate the data[17,19]. On the other hand, species-specific populations may become obscure if the algorithm overfits[16]. Importantly, mapping genes from different species by gene homology can cause significant information loss since homology annotation can be challenging for species without a well-annotated genome or between evolutionarily distant species[30]. The method SAMap tackles this issue by reciprocally and iteratively updating a gene-gene mapping graph from de-novo basic local alignment search tool (BLAST) analysis and a cell-cell mapping graph to stitch atlases between species[19]. SAMap can lay much stronger mapping of homologous cell types across distant species and is capable of discovering gene paralog substitution events. However, SAMap is computationally intensive and designed for whole-body alignment, while methods optimised for large-scale integration may be more scalable to multiple datasets and more species.

It is unclear which strategy for cross-species integration yields the most informative results and how the selection of homologous genes and integration algorithms affect the observable relationships between cells. This is because strategies have not been tested head-to-head across a variety of organs, species and biological systems. Furthermore, the number of species involved in the integration and the time that species have diverged can impact the output characteristics differently among strategies.

In this work, we develop the BENchmarking strateGies for cross-species integrAtion of singLe-cell RNA sequencing data (BENGAL) pipeline. We run BENGAL to examine 28 cross-species integration strategies, covering 4 ways of mapping genes by homology and 10 integration algorithms in 16 tasks. Our analysis focuses on cell types that exhibit a clear one-to-one correspondence across some vertebrate species. Through this, we identify strategies that produce reliable embeddings in this fundamental scenario in various biological contexts. We reason that only when established homologous cell types are well-integrated, the output from the respective approach can then be further explored through subsequent analysis, such as a deeper clustering to identify cell states at a higher granularity or to decipher complex cell type taxonomy. To quantitatively analyse integration results, we use established batch correction metrics and biology conservation metrics to analyse the integration of homologous cell types between species, and the loss of biological heterogeneity, respectively. To address overcorrection, we develop a new biology conservation metric to address loss of cell type distinguishability due to integration. We further perform cross-annotation of cell types using the integrated data to determine if the integration enables cell type annotation transfer. Finally, by observing the behaviour of different integration strategies in tasks with certain examination points, we provide guidelines for appropriate strategy selection aligned to cross-species integration goals.

## Results

### Integration of cross-species scRNA-seq data

The BENGAL pipeline performs cross-species integration and assessment of integration results using 28 strategies, as illustrated in Fig. 1. Quality control (QC) and curation of cell ontology annotations of input data are required prior to running the pipeline and should be performed in an input-specific manner. (see Methods for details and recommended practices). During integration, the pipeline first translates orthologous genes between species using ENSEMBL multiple species comparison tool[31] and concatenates raw count matrices from different species. We compared three approaches for cross-species gene homology mapping: mapping using only one-to-one orthologs; mappings including one-to-many or many-to-many orthologs by selecting those with a high average expression level, or with a strong homology confidence (see Methods). Notably, LIGER UINMF also takes unshared features, therefore genes without annotated homology are added on top of the mapped genes. We fed the concatenated raw count matrix to 9 integration algorithms, some of which were top-performers from previous benchmarking of data integration[17], including fastMNN, Harmony, LIGER, LIGER UINMF, Scanorama, scVI, scANVI, SeuratV4CCA and SeuratV4RPCA. SAMap requires de-novo reciprocal BLAST to construct a gene-gene homology graph and we therefore followed the standalone workflow.

### Output assessment

Integration outputs were assessed from three main aspects, species mixing, biology conservation and annotation transfer. We computed and compared 4 batch correction metrics and 5 biology conservation metrics for species mixing and biology conservation, respectively (Fig. 1, see Methods for metrics selection and applicability, see Supplementary Methods for metrics principle). We also calculated these metrics in 3 types of homology concatenated, unintegrated data, to contextualise the raw scores. Metrics scores in integrated data from 27 strategies (excluding SAMap) and in unintegrated data were min-max scaled per task to equalise their discriminative power (see Methods for scaling detail). The species mixing score is the average of applicable scaled batch correction metrics, while the biology conservation score is the average of applicable scaled biology conservation metrics. The integrated score is a weighted average of species mixing score and biology conservation score with a 40/60 weighting[17]. It is worth noting that batch correction metrics are not applicable to SAMap outputs, so we did not include it in the scoring and ranking (see Supplementary Note 1 for reasons). Instead, we show Uniform Manifold Approximation and Projection (UMAP) for visual inspection and alignment score to quantify the percentage of cross-species neighbours (see Methods for alignment score calculation).

Importantly, we developed a new metric, Accuracy Loss of Cell type Self-projection (ALCS), to target the most unwanted artefact of cross-species integration. ALCS quantifies the degree of blending between cell types per-species after integration, thus indicating the tendency of overcorrection of cross-species heterogeneity that may obscure species-specific cell types. We employed the self-projection concept in machine learning implemented in our previous work Single Cell Clustering Assessment Framework (SCCAF)[32] and used the test accuracy of self-projection to indicate how well cell type labels are distinguishable from each other on a given feature representation. The loss of self-projection accuracy after integration, compared with unintegrated per-species data, measures the loss of cell type distinguishability due to integration (see Methods for further details of ALCS). We report ALCS separately to highlight its importance in our cross-species integration setting.

For annotation transfer, a multinomial logistic classifier built upon the SCCAF framework was trained on one species and used to annotate cell types in another species based on the shared features in the integrated embedding. We assessed annotation transfer by calculating

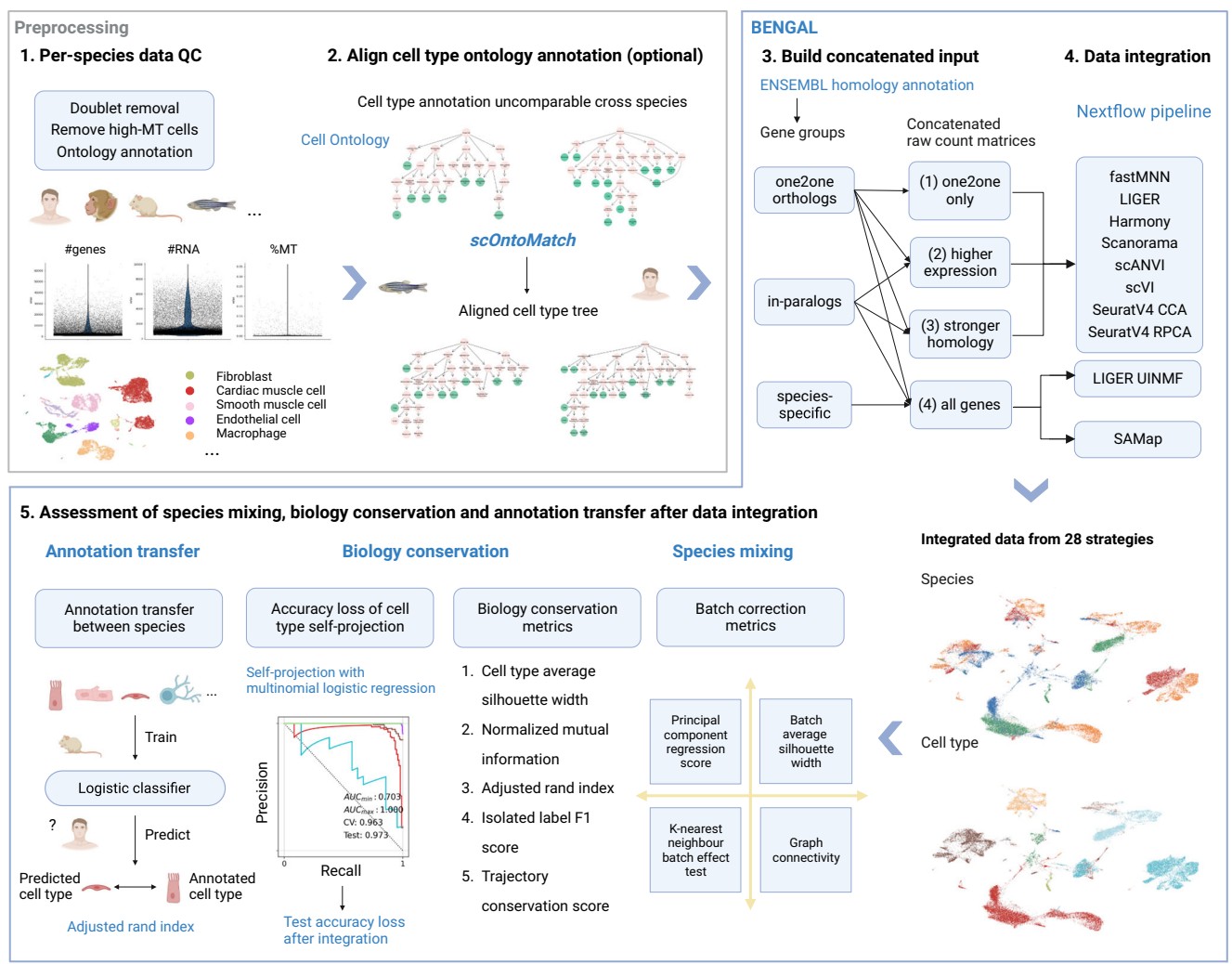

**Fig. 1 | Schematic of the BENGAL pipeline. 1** Quality control of input data is performed prior to data integration. and is not part of BENGAL. Potential doublets and low-quality cells expressing high mitochondrial genes should be removed. Cell ontology annotations are collected from atlases or data portals or curated from the originally published annotation. **2** When the granularity of ontology annotation is incomparable across datasets, one-to-one homology between cell types needs to be robustly aligned. We developed scOntoMatch to find the appropriate annotation granularity and align cell type hierarchies across given datasets (see Methods). **3** Genes are grouped and translated across species by homology defined in ENSEMBL multiple species comparison tool. Raw count matrices are then concatenated across species using four possible homology matching methods respective to method inputs. **4** Run 9 integration algorithms to generate integrated output. **5** Perform integration assessment from species mixing, biology conservation and cell type annotation transfer. BENGAL, BENchmarking strateGies for cross-species integrAtion of singLe-cell RNA sequencing data; QC, quality control; MT, mitochondrial; SCCAF, single cell clustering assessment framework; AUC, area under the curve; CV, cross-validation. Created with BioRender.com.

Adjusted Rand Index (ARI) between the original and transferred annotation (see Methods for details).

### Benchmarking metrics of different integration strategies varied widely

We ran the BENGAL pipeline for 16 cross-species integration tasks spanning a variety of biological scenarios to observe species mixing and biology conservation (see Table 1 for task design and Methods for tasks abbreviation). We explored performance of the different strategies in 3 different adult tissues, namely pancreas, hippocampus and heart, as well as in whole-body embryonic development. Using heart data from 5 species, we explored the upper limit of the number of species to include in one integration with 4 tasks. In addition, we performed pairwise integration to examine the impact of divergent time between species on integration output with 10 tasks.

To provide an overview of strategies' performance, we calculated the mean and standard deviation of integrated scores for all strategies using the 7 reference tasks (excluding pairwise integration tasks as they overlap with the heart tasks, Table 1, Fig. 2a). Overall, major differences were given by integration algorithms but not homology methods. Strategies that achieved successful integrations stayed largely consistent across tasks, while the relative species mixing and biology conservation scores varied. In general, scANVI, scVI, SeuratV4 CCA, SeuratV4 RPCA, harmony and fastMNN performed well across the board. For each task, we analysed strategies' relative species mixing and biology conservation (Fig. 2b). scVI, harmony and SeuratV4 RPCA ranked top3 in terms of average species mixing across tasks, while for biology conservation, it was scANVI, SeuratV4 CCA and SeuratV4 RPCA (Supplementary Data 1). Across all tasks, we observed a trade-off between species mixing and biology conservation in most strategies (Fig. 2b, see Supplementary Figs. 1–16 for detailed metrics and scores of each task). SAMap yield strong cross-species alignment (see alignment score in Supplementary Fig. 17 and UMAP visualisation in Supplementary Figs. 27–42).

**Table 1 | Overview of the reference tasks for benchmarking strategies for cross-species integration of scRNA-seq data**

| Task name | Pancreas_hs_mm[45] | Hippocampus_hs_mu_ss[33] | Embryo_xt_dr[34,35] | Heart_hs_mf[1,3] | Heart_hs_mf_mm[1,3,36] | Heart_hs_mf_mm_xl[1,3,36,46] | Heart_hs_mf_mm_xl_dr[1,3,5,36,46] |
|---|---|---|---|---|---|---|---|
| Species | *Homo sapiens*[a], *Mus musculus* | *Homo sapiens, Macaca mulatta, Sus scrofa* | *Xenopus tropicalis, Danio rerio* | *Homo sapiens, Macaca fascicularis* | *Homo sapiens, Macaca fascicularis, Mus musculus* | *Homo sapiens, Macaca fascicularis, Mus musculus, Xenopus laevis* | *Homo sapiens, Macaca fascicularis, Mus musculus, Xenopus laevis, Danio rerio* |
| Technology | inDrop | snRNA-seq | inDrops | scRNA-seq with 10× Genomics 3' V3.1 (*H.sapiens*), snRNA-seq with DNBelab C Series Single-Cell Library Prep Set (*M.fascicularis*) | same with 4, scRNA-seq with 10× Genomics 3' V2 (*M.musculus*) | same with 5, microwell-seq (*X.laevis*) | same with 6, microwell-seq (*D.rerio*) |
| Number of cells per batch | *Homo sapiens*: 8,402; *Mus musculus*: 1,875 | *Homo sapiens*: 9,170, 8,214, 7,907, 6,893, 6,091, 5,555; *Macaca mulatta*: 19,092, 8,678, 8,337; *Sus scrofa*: 13,168, 12,869, 10,814 | *Xenopus tropicalis*: 123,632; *Danio rerio*: 36,627 | *Homo sapiens*: 12,747; *Macaca fascicularis*: 10,465 | same with 4, *Mus musculus*: 7,402 | same with 5, *Xenopus laevis*: 21,427 | same with 6, *Danio rerio*: 3,193 |
| Number of analysed O2O homologs | 11,248 | 12,557 | 5188 | 12,998 | 11,202 | 6609 | 4267 |
| Number of analysed O2M and M2M homologs | 664 | 24 | 1749 | 398 | 199 | 99 | 82 |
| Challenge presented | Basic performance | Large data size, complex cell type structure | Whole-body data, challenging homology, developmental trajectory | Cross-study and cross-technology integration | Cross-study and cross-technology integration | Evolutionarily distant species, challenging homology | Evolutionarily distant species, challenging homology |

The "Number of cells per batch" refers to the cells from each batch that have passed the quality control criteria in the original literature and were included in the integration. The "Number of analysed O2O homologs" and "Number of analysed O2M and M2M homologs" pertain to the genes that were analysed in the integration tasks satisfying the following criteria: (1) quantified in all datasets involved, (2) can be mapped to an ENSEMBL gene ID, and (3) have homology annotation across all studied species. To identify one-to-one orthologs and in-paralogs, including one-to-many and many-to-many orthologs, we used the ENSEMBL multiple species comparison tool (version 106) and accessed it via biomaRt (v2.46.3). O2O, one-to-one; O2M, one-to-many; M2M, many-to-many.

[a]The scientific names of species are in italic formatting.

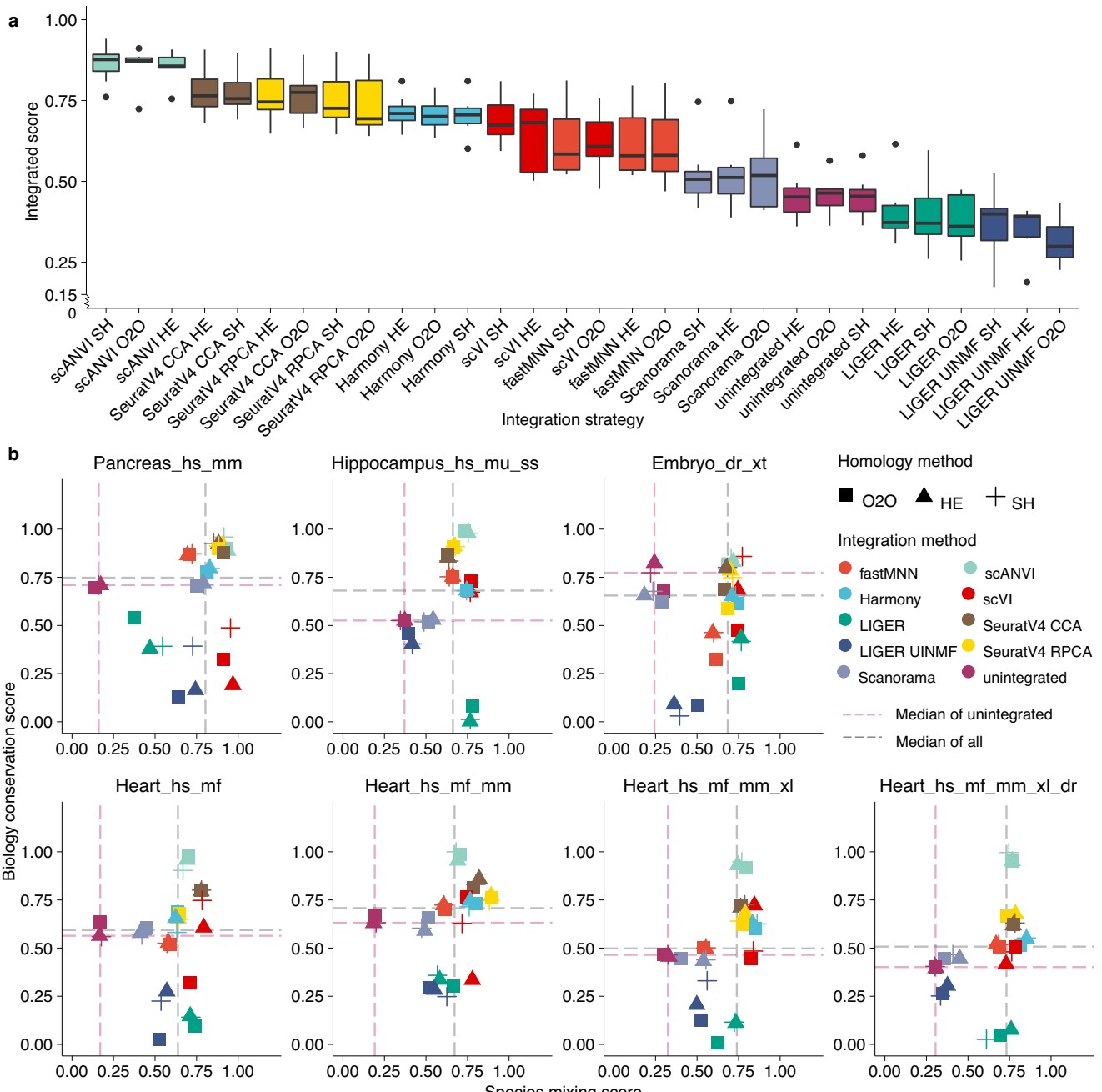

**Fig. 2 | Benchmarking scores and metrics of different integration strategies on cross-species analysis. a** We used 4 batch correction metrics and 5 biology conservation metrics to examine the output of 28 data integration strategies and rank the strategies based on 7 cross-species data integration tasks (see Methods for metrics and tasks details). Batch correction metrics and biological conservation metrics are min-max scaled per task. The species mixing score is the average of scaled batch correction metrics applicable to the output type of the integration algorithm, while the biological conservation score is the average of scaled applicable biology conservation metrics to the output type and to the integration task (see Methods for metrics and scaling details). The integrated score is a weighted average of species mixing score and biology conservation score with 40/60 weight, respectively. The bar in the boxplot shows the arithmetic mean, lower and upper

hinges correspond to the first and third quartiles, whiskers extend from the hinge to the largest value no further than 1.5 * interquartile range and outliers beyond this range are plotted as individual data points. **b** The species mixing score and biology conservation score of 7 example tasks (see Table 1 for task details and Methods for task naming). O2O, only use one-to-one orthologs; HE, one-to-one orthologs plus one-to-many and many-to-many orthologs matched by higher average expression level; SH, one-to-one orthologs plus one-to-many and many-to-many orthologs matched by stronger homology confidence. hs, *Homo sapiens*, human; mf, *Macaca fascicularis*, long-tailed macaque; mu, *Macaca mulatta*, rhesus macaque; mm, *Mus musculus*, mouse; ss, *Sus scrofa*, pig; xl, *Xenopus laevis*, African clawed frog; xt, *Xenopus tropicalis*, Western clawed frog; dr, *Danio rerio*, zebrafish.

## Loss of cell type distinguishability after integration

After examining output characteristics using established metrics for species mixing and biology conservation, we calculated ALCS to observe the conservation of cell type distinguishability. Figure 3a shows the ALCS of various strategies in 7 example tasks, ordered by

increasing maximum divergence time among integrated species. We found that as the integration involves more species that have diverged for a longer time, there is a general increase in ALCS across all strategies, but to a different extent. Overall, LIGER, LIGER UINMF and fastMNN have noticeably higher ALCS. As an example, we show

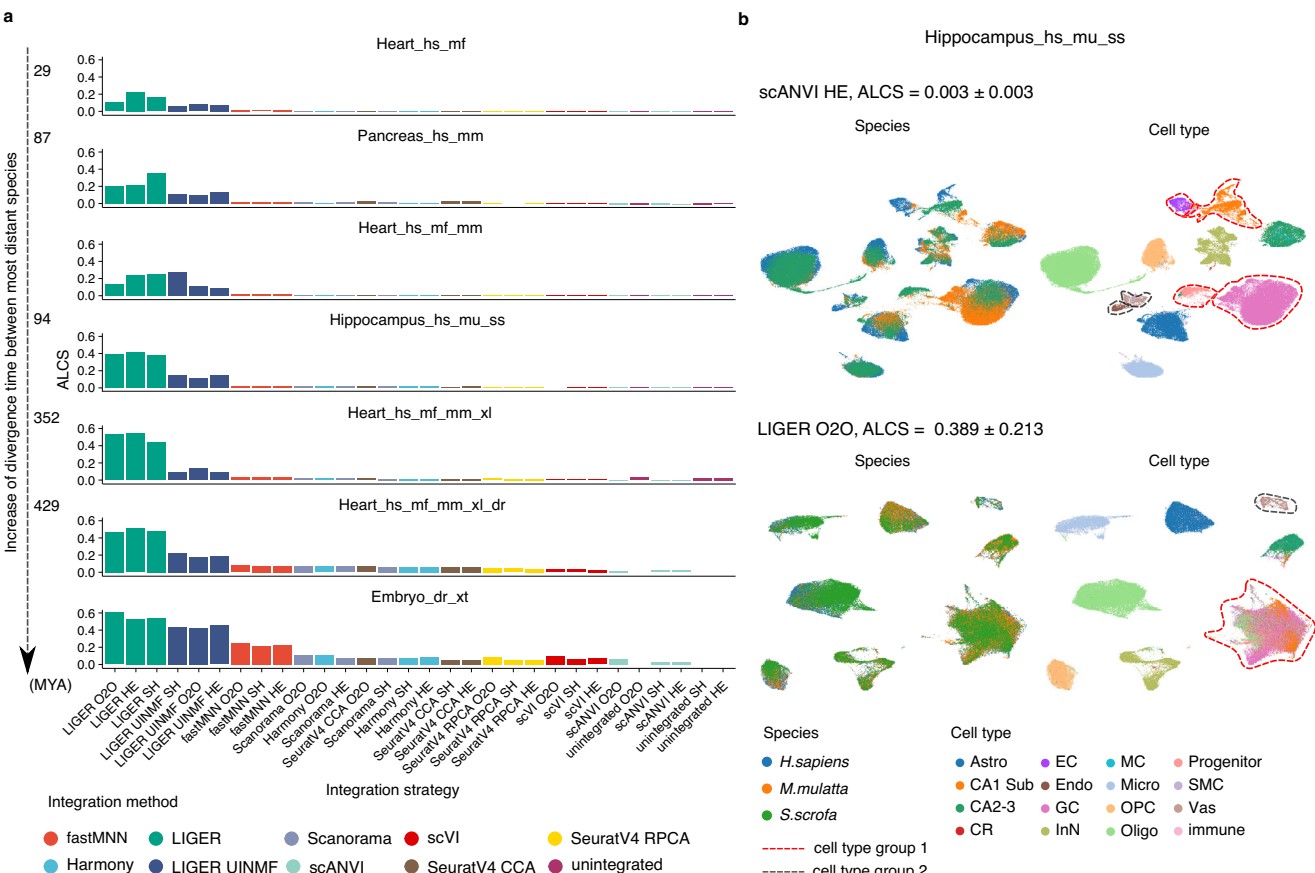

**Fig. 3 | Loss of cell type distinguishability after integration is measured by accuracy loss of cell type self-projection. a** The ALCS of different integration strategies in 7 cross-species integration tasks. ALCS is the decrease in test accuracy of a cell type classifier trained on integrated data, compared with unintegrated data (see Methods for details). A high ALCS indicates that cell types become indistinguishable from each other after the integration and suggests overcorrection. Tasks are ordered by increasing divergent time between the most distant species. **b** Example of strategies with high and low ALCS in the Hippocampus_hs_mu_ss task. In scANVI HE integrated data, all cell types are clearly distinguishable after integration. However, In LIGER O2O results, EC, CA1 Sub, GC and some Oligo (cell type group **1**), as well as Endo with Vas (cell type group **2**) became merged. ALCS, accuracy loss of cell type self-projection; MYA, million years ago; O2O, only use one-to-one orthologs; HE, one-to-one orthologs plus one-to-many and many-to-many orthologs matched by higher average expression level; SH, one-to-one orthologs plus one-to-many and many-to-many orthologs matched by stronger homology confidence; hs, *Homo sapiens*, human; mf, *Macaca fascicularis*, long-tailed macaque; mu, *Macaca mulatta*, rhesus macaque; mm, *Mus musculus*, mouse; ss, *Sus scrofa*, pig; xl, *Xenopus laevis*, African clawed frog; xt, *Xenopus tropicalis*, Western clawed frog; dr, *Danio rerio*, zebrafish. Astro, astrocytes; CR, Cajal-Retzius cells; EC, entorhinal cortex; Endo, endothelial cells; GC, granule cell; InN, inhibitory neurons; MC, mossy cell; Micro, microglia cells; OPC, oligodendrocyte progenitor cells; Oligo, oligodendrocytes; NP, neuronal progenitors; SMC, smooth muscle cells; Vas, vasculature.

the Uniform Manifold Approximation and Projection (UMAP) visualisation of scANVI HE and LIGER O2O integration of the hippocampus_hs_mu_ss task (Fig. 3b). Though both methods were capable of species mixing, LIGER merged several groups of cell types, for instance among EC, CA1 Sub, GC and Oligo (cell type group 1, red dashed circle), as well as Endo with Vas (cell type group 2, dark grey dashed circle), whereas these cell types were kept distinguishable in scANVI. The relative ALCS stayed consistent in the 10 pairwise integration strategies of species in the heart tasks, where we also observed higher ALCS between species pairs that have diverged for a longer time (Supplementary Fig. 18).

We conclude that cell type distinguishability is in general at risk when integrating evolutionarily distant species, but scANVI and scVI can produce more robust results than others. When the goal of cross-species integration is to uncover cell type homology, it is important to choose methods with low ALCS, to derive correct interpretations of cell type similarity based on the integrated embedding.

### Integration of pancreas and hippocampal-entorhinal system

We demonstrate BENGAL, starting from the integration between human and mouse pancreas (Pancreas_hs_mm task, Fig. 4a–c), and human, rhesus macaque and pig hippocampal-entorhinal system[33] (Hippocampus_hs_mu_ss task, Fig. 4d–f). These two scenarios have clearly distinguishable cell types and evident one-to-one cell type homology. The data were generated in the same study thus reducing technical batch effects. In comparison with the Pancreas_hs_mm task, the Hippocampus_hs_mu_ss task challenged the strategies with a larger data size and more complex subcluster structures (Table 1).

Prior to integration, cells clustered strongly by species of origin (Fig. 4b, e). After integration and assessment, we observed diverse outputs from different strategies (Fig. 4c, f, Supplementary Figs. 1, 2). Overall, the integrated scores ranged between 0.33–0.94 for the Pancreas_hs_mm task or 0.31–0.89 for the Hippocampus_hs_mu_ss task. Batch correction scores were between 0.14 and 0.97 or 0.35 and 0.78, and biology conservation scores between 0.13 and 0.96, or 0 and 0.99, for the pancreas or Hippocampus_hs_mu_ss task, respectively. In both cases, harmony, fastMNN, scANVI, scVI, SeuratV4 CCA, SeuratV4 RPCA and SAMap were able to integrate the data while scanorama, LIGER and LIGER UINMF did not integrate some homologous cell type pairs, such as beta cells and delta cells (Supplementary Fig. 31). Subcluster structures were visible on the UMAPs by scVI/scANVI but not on outputs by

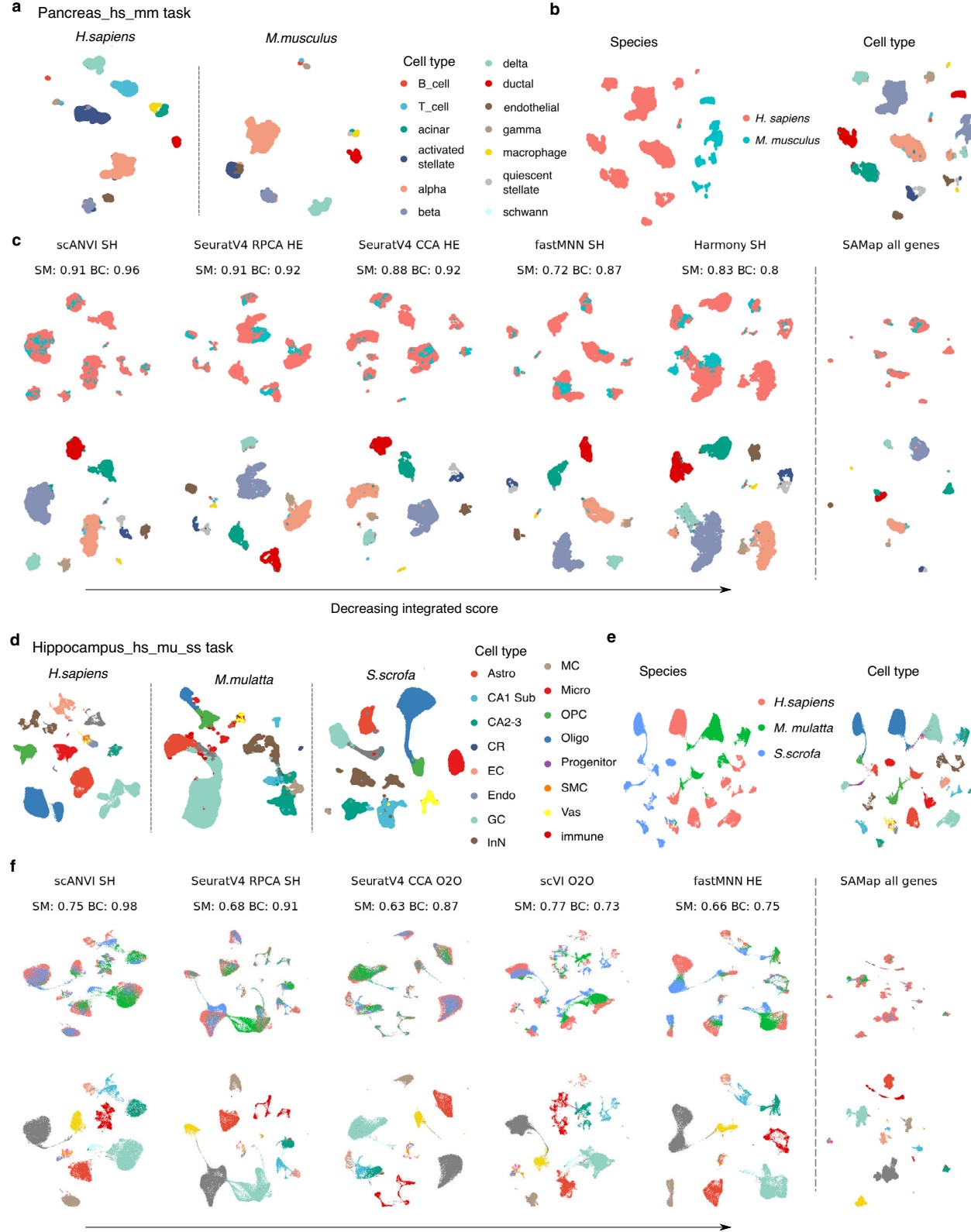

other methods in the Hippocampus_hs_mu_ss task (Fig. 4f). We hypothesize that heterogeneity within each cell type was better preserved in these deep learning-based methods, compared with nearest neighbour-based methods. SAMap gave a strong cross-species alignment, leading to a high degree of overlap between shared cell types on the UMAPs.

**Integrating embryonic development**

Next, we integrated the whole-body embryonic development process between xenopus and zebrafish. This is more challenging than integrating mature tissues because cell types are less distinct and a continuous trajectory exists among cells from different developmental stages. Based on mapped homologous cell types and recorded hours

**Fig. 4 | Integration results of human and mouse pancreas and human, rhesus macaque and pig hippocampal-entorhinal system. a** UMAP visualisation of cell types in human and mouse pancreatic tasks. **b** UMAP visualisation of unintegrated data of the pancreas task. In unintegrated data, cells cluster primarily by species and homologous cell types form different species from separate clusters. One-to-one orthologs between human and mouse were used to concatenate raw count matrices, then processed by standard analysis pipeline (see Methods). **c** UMAP visualisation of integration results of five top-performing algorithms and SAMap of the pancreas task, organised by decreasing integrated score and coloured by species or cell type. All strategies were ranked by integrated score and for the top 5 integration algorithms, the homology strategy with the highest integrated score was shown. The respective species mixing and biology conservation scores were also shown. The scores were not calculated for SAMap (not applicable, see Methods

for details). **d–f** UMAP visualisation of original data, unintegrated data and integration results of five top-performing strategies and SAMap of the hippocampus task. SM, species mixing score; BC, biology conservation score; O2O, only use one-to-one orthologs; HE, one-to-one orthologs plus one-to-many and many-to-many orthologs matched by higher average expression level; SH, one-to-one orthologs plus one-to-many and many-to-many orthologs matched by stronger homology confidence; hs, *Homo sapiens*, human; mu, *Macaca mulatta*, rhesus macaque; mm, *Mus musculus*, mouse; ss, *Sus scrofa*, pig; Astro, astrocytes; CR, Cajal-Retzius cells; EC, entorhinal cortex; Endo, endothelial cells; GC, granule cell; InN, inhibitory neurons; MC, mossy cell; Micro, microglia cells; OPC, oligodendrocyte progenitor cells; Oligo, oligodendrocytes; NP, neuronal progenitors; SMC, smooth muscle cells; Vas, vasculature.

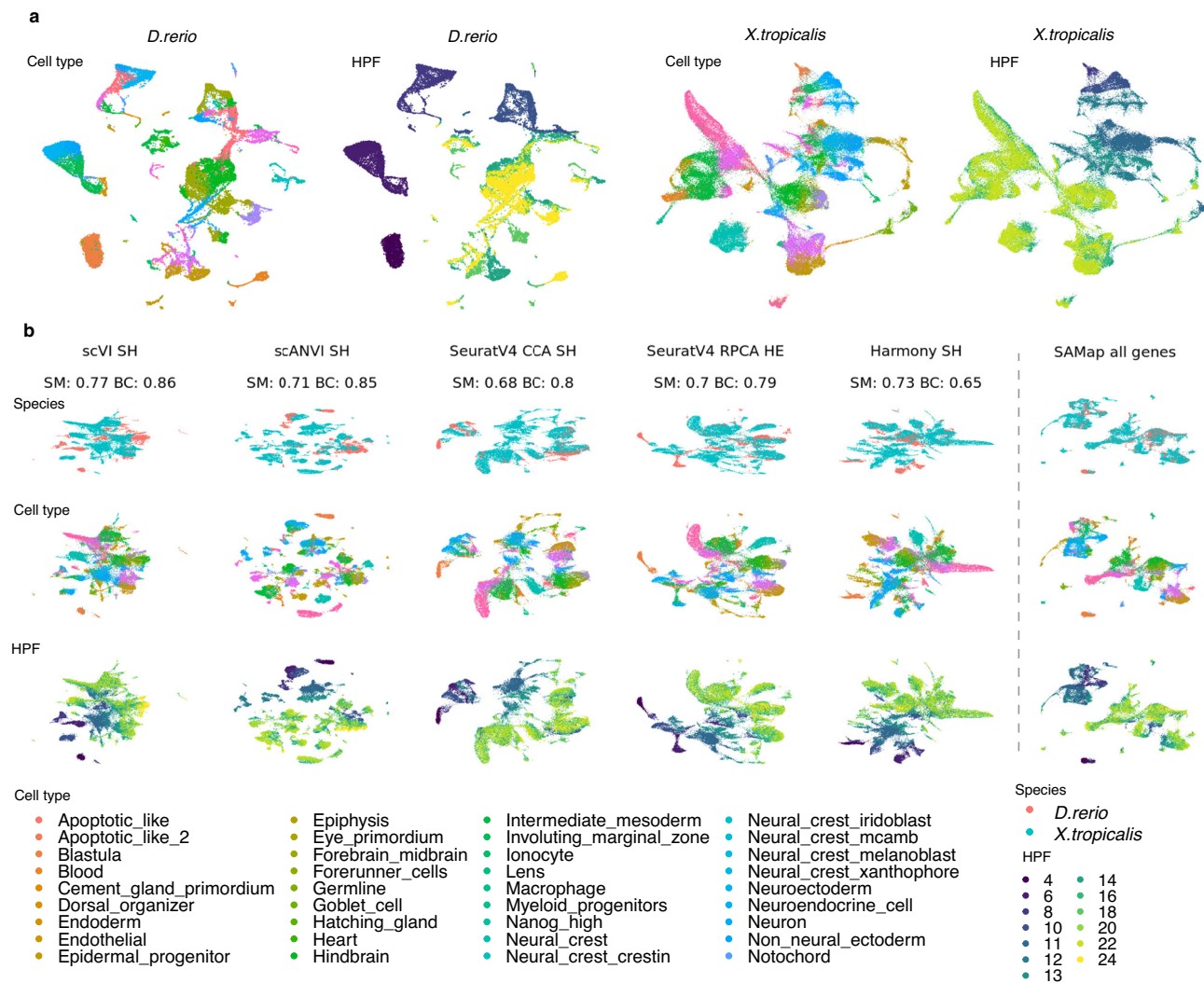

**Fig. 5 | Integration of embryonic development between xenopus and zebrafish.**
**a** UMAP visualisation of unintegrated data of xenopus and zebrafish embryonic development. Cells are coloured by their cell types and the developmental stages as in HPF. **b** UMAP visualisation of integration results of five top-performing algorithms and SAMap, organised by decreasing integrated score and coloured by species or cell type. Note that 5 types of Neural_crest cells in zebrafish were

combined into Neural_crest in the integration for matching with xenopus. All strategies were ranked by integrated score and for the top 5 integration algorithms, the homology strategy with the highest integrated score was shown. The respective species mixing and biology conservation scores were also shown. The scores were not calculated for SAMap (not applicable, see Methods for details). SM, species mixing score; BC, biology conservation score; HPF, hour post fertilisation.

post fertilisation (HPF) between a zebrafish and xenopus embryonic development dataset[34,35], we examined species mixing and biology conservation of various integration outputs (Fig. 5a, b).

We observed an overall lower rate of success across strategies in the Embryo_dr_xt task. Algorithms that achieved higher integrated scores than unintegrated data include scVI/scANVI, SeuratV4 methods

and Harmony, and they had comparable species mixing and biology conservation scores (Fig. 5b). However, due to the homology matching process, maximum 6937 homologous genes among 17,330 and 9661 genes were included in the analysis for zebrafish and xenopus, respectively, while SAMap were able to include all genes due to its utilisation of a gene-to-gene BLAST graph. Hence, when integrating

species with less well-annotated gene homology, SAMap can keep a much more complete profile of cellular expression when successfully integrating the data. On the other hand, we found that SAMap had a relatively low trajectory conservation score in SAMap (Supplementary Fig. 19). As seen on the SAMap UMAP, homologous cell types were strongly linked to each other and cells from different developmental stages were partially mixed. However, methods such as scVI/scANVI and SeuratV4 displayed a more explicit progression of developmental stages along the embedding (Fig. 5b). This observation suggests that SAMap focuses on prioritising identity-defining expression characteristics that stitch together evolutionarily related cell types.

### Cross-atlas integration of heart and aorta among five species

We designed a challenging task that is to integrate the heart and aorta tissue from five species: human, long-tail macaque, mouse, xenopus and zebrafish. While the mouse data were from an independent study to cover more cell types[36], all other species data were from current multi-organ atlases. We performed the integration by gradually adding one more species, starting from human and long-tail macaque, to explore the upper limit of the number of species to generate informative integration results. We have also performed pairwise integration of all species to examine the impact of divergence time on integration results.

When comparing cell type annotations between atlases, we noticed that the annotation granularity varied greatly (Supplementary Fig. 20). To robustly match homologous populations between atlases based on existing expert annotation, we employed the cell ontology (CL), a species-neutral, controlled vocabulary system that describes the ancestor-descendant relationship of cell type terms. CL is the current standard vocabulary used in both the Human Cell Atlas[37] and the EBI Single Cell Expression Atlas[38] for cell type annotation. Leveraging the hierarchical organisation of CL, we developed scOntoMatch, an R package which aligns the granularity of ontology annotations among scRNA-seq datasets to make them comparable across studies (see Methods for details of the scOntoMatch algorithm). Applying scOntoMatch to the heart data, we found the highest common granularity for cell type annotation in different integration runs.

After integration, scANVI reached the highest integrated score (0.62–0.69, Fig. 6a, individual metrics unscaled for each task for cross-task comparison. The same applies to the scores below). This was largely due to its high biology conservation score (0.70–0.83). On the other hand, SeuratV4 RPCA methods gave the highest species mixing score for the four tasks (0.46–0.54). For all tasks, scanorama, LIGER and LIGER UINMF had lower integrated scores than unintegrated data. In general, addition of species did not affect the relative performance of the integration algorithms and the species mixing scores, highlighting their robustness by these criteria (Fig. 6a). In contrast, biology conservation scores and ALCS dropped significantly (Figs. 6a, 3a), suggesting a general loss of biological signals of cell type specificities when distant species were added to the integration.

Since only 8 among 22 cell types were shared by all species, we took this opportunity to investigate how different strategies handle cell types found in few datasets using isolated label F1 (iso_F1) score. We found that nearest neighbour-based methods have a tendency to overcorrect, leading to merging of similar cell types, especially when more species are involved in the integration (Fig. 6c). As an example, we show the SeuratV4 CCA integration of one-to-one orthologs only data for the four tasks (Fig. 6d, e). Among methods that achieved integration, SeuratV4 CCA had a low iso_F1 score, suggesting cell types specific to few species data were not well-separated (Supplementary Figs. 4–7).

Increase of ALCS with addition of species was broadly observed for most strategies, indicating a loss of cell type distinction when many species data were involved in one integration (Supplementary Fig. 21). For example, ALCS of SeuratV4 CCA O2O was roughly maintained

between integration of human and macaque or human, macaque and mouse, but adding xenopus to the integration led to some loss of cell type distinction and this became global and stronger after zebrafish was integrated (Fig. 6d). Integrating many species that span a large evolutionary distance cannot therefore yield robust results for downstream analysis, such as de novo clustering, due to information loss during gene homology mapping and the integration process. Similarly, pairwise integration also suggested that integration is more prone to loss of cell type distinction between species that have diverged for a longer time (Supplementary Fig. 18).

### Cell type annotation transfer cross-species

One key application of cross-species integration is cell type annotation transfer: using annotated cell types in a well-studied species to label the cell types of a newly characterised species. To investigate the potential of different strategies to support annotation transfer, we performed pairwise cross-annotation in all tasks. In practice, we applied the SCCAF multinomial logistic classifier trained on one species to predict the cell type labels of another species. When the transferred labels covered all known sharded cell types and had an acceptable adjusted rand score (ARI) with the original label, we considered cross-annotation successful.

Overall, we observed that if data integration was well-achieved, annotation transfer was mostly successful (Fig. 7). In all tasks, strategies showed a significant positive correlation of integrated score with average ARI (Spearman's rank correlation $P$-value $< 0.001$). Compared with species mixing score, biology conservation score correlated stronger with ARI, suggesting that cross-annotation is facilitated by well-preserved cell type features (Supplementary Fig. 22). We noticed that with the addition of evolutionarily more distant species in the heart tasks, the overall annotation transfer quality did not drop for methods that achieved successful integration (Fig. 7). However, when integration is not achieved, annotation transfer can result in a negative ARI, meaning that the process is more erroneous than random label assignment. Inspection of UMAP visualisation of showed that transferred annotation was to a large extent accurate in methods such as scANVI and SeuratV4 CCA even between distant species, such as human and zebrafish (see Supplementary Fig. 23 for successful and unsuccessful examples). This is in line with the observation that top-performing methods stayed relatively robust with the addition of more species (Fig. 6a). Pairwise integration tasks show similar results that annotation transfer is more difficult between evolutionarily distant species for all strategies (Supplementary Fig. 24).

### Impact of one-to-many and many-to-many homologous genes

We found that incorporating one-to-many and many-to-many homologous genes was beneficial when they accounted for a high proportion of genes used in the integration. Specifically, in the Embryo_dr_xt task, algorithms which successfully integrated the data achieved higher integrated scores if in-paralogs were included, especially when the in-paralog with higher homology confidence was matched with the corresponding ortholog (Supplementary Fig. 25). The improvement was primarily driven by increased biology conservation scores, suggesting that including these genes preserved a more complete picture of cell type expression profiles which facilitated data integration. This observation is consistent with SAMap outperforming in the Embryo_dr_xt task, as SAMap used all genes and was better able to resolve the alignment between homologous cell types (Fig. 5, Supplementary Fig. 17). Although we did not observe significant improvements in benchmarking scores by adding in-paralogs in most tasks, we found that the homology method still influenced integration from the perspective of highly variable genes (HVGs). We observed that different HVGs were selected, especially among evolutionarily distant species, when incorporating in-paralogs (Supplementary Fig. 26). Since

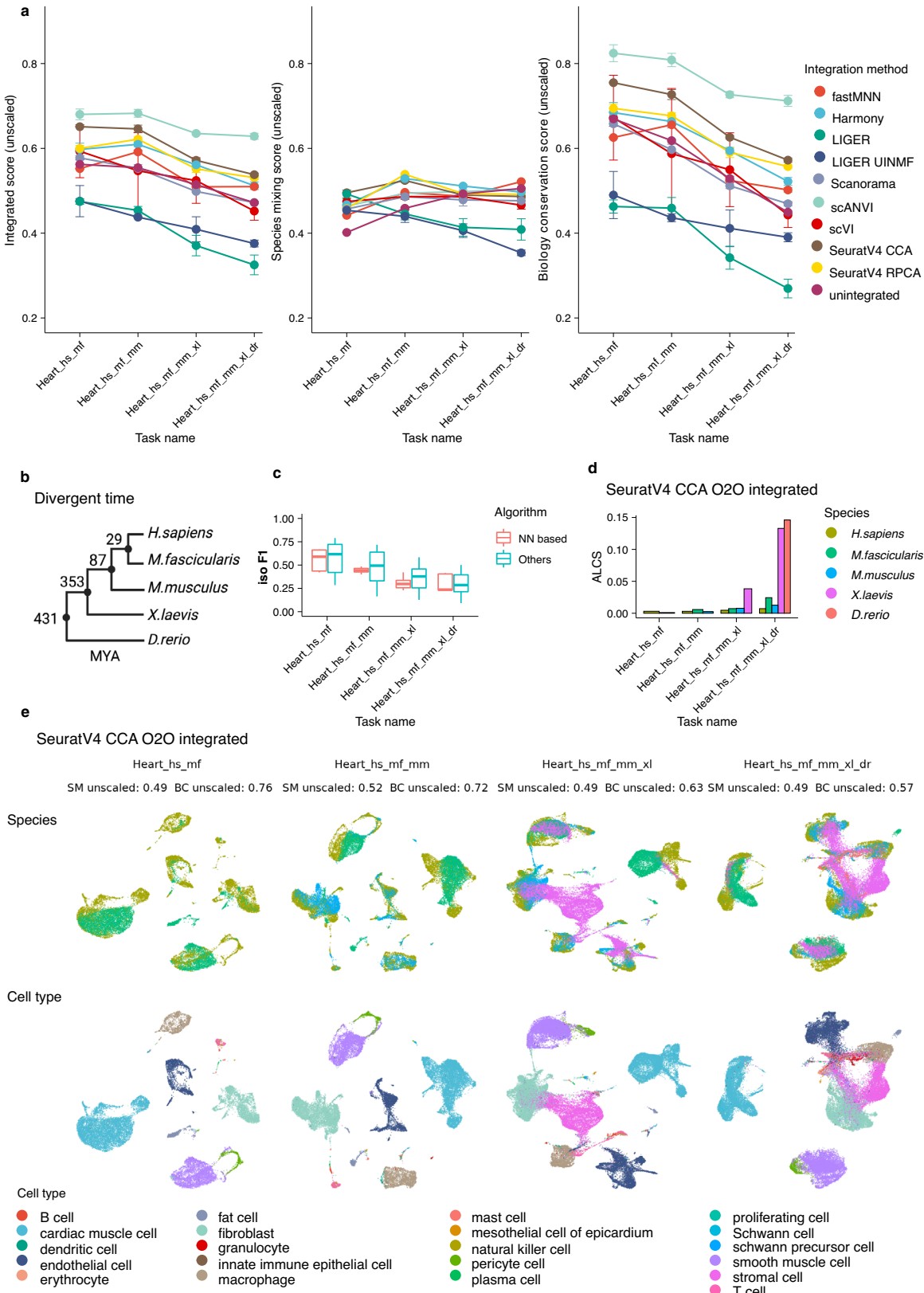

integration algorithms other than SAMap require HVG selection, adding in-paralogs that contribute to cell type expression variation which would support the integration algorithms. Overall, our findings highlight the importance of carefully considering the inclusion of in-paralogs in data integration, as their impact varies depending on the specific task and species being studied.

## Discussion

We benchmarked 28 strategies for cross-species single-cell transcriptomics data integration, covering 4 ways to match genes cross-species by homology and 10 integration algorithms. Based on consensus cell type homology and unified ontology annotation, we used 4 metrics to evaluate the degree of species mixing and 6 for scoring

**Fig. 6 | Integrating heart tissue from human, long-tail macaque, mouse, xenopus and zebrafish. a** Unscaled integrated score, species mixing score and biology conservation score of all strategies for the four heart tasks. The batch metrics and biology conservation metrics are not min-max scaled per task to enable cross-task comparison. Each point shows the average score across three homologous methods of each integration algorithm and error bars indicate the standard deviation. **b** The divergence time among the studied species in millions of years. **c** The isolated label F1 score of different strategies in the 4 heart tasks. Nearest neighbour-based (NN-based) results (*n* = 9) are by algorithms including fastMNN, SeuratV4 CCA and SeuratV4 RPCA and the other results (*n* = 18) are by non-NN-based algorithms. The bar in the boxplot shows the arithmetic mean, lower and

upper hinges correspond to the first and third quartiles and whiskers extend from the hinge to the largest value no further than 1.5 * interquartile range from the hinge. There are no outliers. **d** The ALCS of SeuratV4 CCA O2O integrated data per species of each heart task. High ALCS indicates a strong loss of cell type distinguishability due to overcorrection (see Methods for details). **e** UMAP visualisation of SeuratV4 CCA O2O integrated data in heart tasks, coloured by species and cell type. Unscaled SM and BC scores are also shown. Iso F1, isolated label F1 score; NN, nearest neighbour; ALCS, accuracy loss of cell type self-projection; SM, species mixing score; BC, biology conservation score; O2O, only use one-to-one orthologs; hs, *Homo sapiens*, human; mf, *Macaca fascicularis*, long-tailed macaque; mm, *Mus musculus*, mouse; xl, *Xenopus laevis*, African clawed frog; dr, *Danio rerio*, zebrafish.

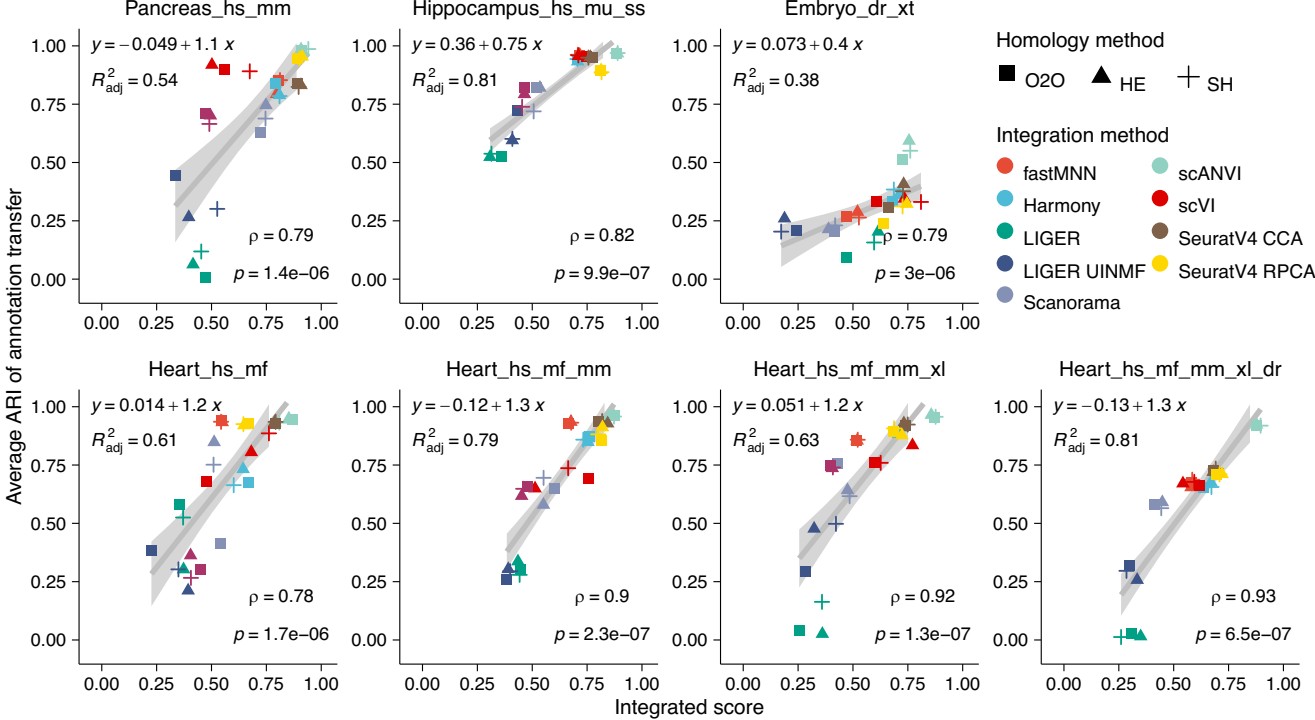

**Fig. 7 | Adjusted rand index of cross-species cell type annotation transfer.** The average ARIs between transferred annotation and original annotation in 7 reference tasks are shown for all integration strategies. A high ARI suggests a successful annotation transfer. Spearman's rank correlation coefficients indicate that ARI significantly positively correlates with the integrated score in all tasks (two-sided Student's *t*-test *P*-value < 0.001). This is further demonstrated by a linear regression using *y* = *x* indicated by the grey line and the band shows 95% confidence interval. Strategies are represented by points whose colour is the integration algorithm and whose shape is the homology method. ARI, adjusted rand index; ρ, Spearman's rank

correlation coefficient; p, *P*-value of Spearman's rank correlation; R²ₐdⱼ, adjusted goodness-of-fit of the linear model; O2O, only use one-to-one orthologs; HE, one-to-one orthologs plus one-to-many and many-to-many orthologs matched by higher average expression level; SH, one-to-one orthologs plus one-to-many and many-to-many orthologs matched by stronger homology confidence; hs, *Homo sapiens*, human; mf, *Macaca fascicularis*, long-tailed macaque; mu, *Macaca mulatta*, rhesus macaque; mm, *Mus musculus*, mouse; ss, *Sus scrofa*, pig; xl, *Xenopus laevis*, African clawed frog; xt, *Xenopus tropicalis*, Western clawed frog; dr, *Danio rerio*, zebrafish.

biology conservation. We also provided the following freely available tools: BENGAL, a Nextflow pipeline for cross-species scRNA-seq data integration and assessment of integration results; ALCS, a cross-species integration-focused biology conservation metric to quantify the loss of cell type distinguishability; scOntoMatch; an R package to help align the granularity of cell ontology annotation across datasets.

We found that deep neural network-based scVI/scANVI generally achieved an overall balance between species mixing and biology conservation. scANVI performed the best across the board, suggesting that taking cell type annotation into account is beneficial for the integration. On the other hand, SeuratV4 methods were able to integrate successfully for evolutionarily distant species but were more prone to overcorrection due to its nearest neighbour foundations. Algorithms based on NMF could not integrate some cell types

and merged different cell types in basic scenarios, suggesting that they were less suitable for cross-species integration. We noticed that the tailor-made cross-species integration algorithm, SAMap, can pick up key similarities between homologous cell types that are otherwise difficult to detect in complex situations. The impact of the homology method depended on the species involved in the integration. Different homology methods influenced HVG selection and had larger influences when there was a high proportion of one-to-many and many-to-many orthologs between the studied species. By performing cell type annotation transfer using integrated results, we observed that if integration is well-achieved, annotation transfer was to a large extent successful even between evolutionarily distant species such as human and zebrafish. Nonetheless, highly similar cell types could still be mislabelled between closely related species *e.g.* human and mouse.

Based on a diversity of cross-species integration scenarios investigated in this study, we provide the following guidelines on choosing the most appropriate algorithm for cross-species integration: for closely related species, scVI (scANVI when confident cell type annotation is available) or harmony carries out species mixing while preserving biological heterogeneity. For relatively distant species, SeuratV4 methods can achieve strong species mixing and RPCA is more scalable than CCA for larger datasets[17,39]. For integration of whole-body atlases or between species lacking well-curated gene homology annotation, SAMap excel in aligning homologous cell types by resolving the gene homology mapping challenge. For species sharing a large amount of one-to-many and many-to-many orthologs, including them in the analysis can improve the integration as more information about cell type expression profiles are preserved. It is important to note that the actual integration of scRNA-seq data is most beneficial among species that have diverged up to a certain extent, such as those from the same phylum. According to the heart example, we conclude that when data integration is performed between non-mammals with mammals, the result can still serve as basis for cell type annotation transfer, but the embedding is not suitable for de novo clustering analysis due to strong loss of biology. Alternative approaches such as correlation analysis of cell type marker genes might be more appropriate for very distant species[16,30].

Our quantitative scoring system of species mixing and biology conservation is supported by two assumptions: (1) cell type annotations are accurate; (2) known cell type homology from literature is confident. It is important to point out that our knowledge about cell types and their evolution is still rapidly growing[8,40]. The increasing availability of scRNA-seq atlases from different species will provide the opportunity to study the evolution of cell types at a greater extent in the future. This study focuses on addressing the computational artefacts introduced by scRNA-seq integration strategies in cross-species scenarios and the different limitations of such approaches, using one-to-one cell type mappings. While valid non-one-to-one cell type mappings across species do exist, particularly over long evolutionary distances[19,30], we are currently limited by: the availability of datasets that can serve as ground truth; sophisticated computational methods; and more specialised assessment metrics to benchmark non-one-to-one evolutionary cell type mappings (see Supplementary Note 2 for further discussions).

Future cross-species integration will require the development of novel integration strategies that overcome the limitation of gene orthology, are aware of the evolutionary homology between cell types and adopt a hierarchical structure of cell type classification. Currently, cross-species integration still heavily relies on curated gene orthology based on sequence. From the standpoint of gene evolution, this has two caveats: (1) species-specific genes contribute to a large extent the novelty of species-specific cell types and should not be ignored when performing cross-species analysis[11,41], (2) functional diversification between orthologs might be as common as between paralogs and out-paralogs can exhibit higher expression similarity compared to their corresponding orthologs through gain, loss and partition of function[19,42,43]. Up till now, only SAMap tackles the above caveats by de novo gene homology mapping and iterative refinement. However, being a heuristic approach, it does not provide explanations of the observed cell type expression similarity. Expression similarities between cell cross-species may arise for reasons other than evolutionary homology, such as convergence and concerted evolution[8]. Novel algorithms are required to disentangle the source of the expression similarities to achieve deeper understandings of cell types[10]. Computational methods able to combine other modalities, such as chromatin accessibility data from single-cell sequencing assay for transposase-accessible chromatin (scATAC-seq) with scRNA-seq data can identify cell type mapping due to evolutionary relatedness. Moreover, as cell types have undergone complex evolution of identity-

defining transcriptional regulatory machinery[19], the correspondence between cell types from different species will not be comprehensively described by a flat mapping such as one-to-one or one-to-many. Adopting a hierarchical representation of cells in the integration outcome can benefit the comprehension of the relationship between cell types across species.

# Methods

## Abbreviations

We used consistent abbreviations for species and homology methods across this study for clarity and brevity. For species, we took the initials of the scientific name of the species: hs, *Homo sapiens*, human; mf, *Macaca fascicularis*, long-tailed macaque; mu, *Macaca mulatta*, rhesus macaque; mm, *Mus musculus*, mouse; ss, *Sus scrofa*, pig; xl, *Xenopus laevis*, African clawed frog; xt, *Xenopus tropicalis*, Western clawed frog; dr, *Danio rerio*, zebrafish. For homology methods the abbreviations were: matching only one-to-one orthologs between species, O2O; matching one-to-one orthologs and in-paralogs with which have a higher expression value, HE; matching one-to-one orthologs and in-paralogs which have a higher homology confidence, SH; include all homologous genes as well as those without homology information, all genes. We named the integration tasks following the pattern: tissue_species, and the integration strategies following the pattern algorithm_homology_method.

## Quality control of scRNA-seq datasets

We followed the QC criteria in the original studies for each dataset. We excluded cells that didn't pass QC from the raw count data, according to published cell type annotations. We did not apply additional QC in this study nor is the QC step part of the BENGAL pipeline, since the criteria are highly specific to each dataset and should be performed on case-by-case while observing the data.

## Reference transcriptomes

Transcriptomes used in the BLAST step in SAMap were downloaded from ENSEMBL, except that for xt in Embryo_dr_xt task the transcriptome was downloaded from Xenbase (https://www.xenbase.org/entry/static-xenbase/ftpDatafiles.jsp). Transcriptome versions were in line with the version at original publication of each dataset: Pancreas_hs_mm task: mm (GRCm38) and hs (GRCh38); Hippocampus_hs_mu_ss task: hs (GRCh38), mu (Mmul_10), ss (Sscrofa11); heart tasks: hs (GRCh38), ma (Macaca_fascicularis_6.0), mm (GRCm39), xl (v9.2), dr (GRCz11); Embryo_dr_xt task: xt (Xtropicalisv9.0), dr (GRCz10).

## Curation of cell type annotation based on known homology

We took the published cell type annotation of all datasets used in this study and curated the cell type annotations into cell ontology[37] terms. We paid specific attention that the same cell type identifier refers to homologous cell types *i.e.* cell types from different species which trace back to the same precursor cell type in a common evolutionary ancestor[8].

In the Pancreas_hs_mm and Hippocampus_hs_mu_ss tasks, cell type homology was evident between species and were matched in the original study. For the Hippocampus_hs_mu_ss task, NBs, RGLs and nIPCs from macaque and pig were collectively annotated as neuronal progenitor. In the human data, we collectively annotated CA1 and SUB as CA1 SUB, CA2, CA3 as CA 2-3, macrophage, microglia, myeloid cell and T cell as immune, aSMCs and vSMCs as smooth muscle cell, pericytes and vascular and leptomeningeal cell as vasculature, COP was combined with OPC. For the Embryo_dr_xt task, we used the annotations provided in the SAMap study[19], as they had manually matched one-to-one homologous cell types using multiple lines of evidence including developmental hierarchies. For the heart tasks, we leveraged the hierarchical structure of cell ontology to align

annotation granularity across datasets, so that homologous cell populations for a given set of species with curated ontology annotation were matched (see the "Aligning ontology annotation with scOntoMatch" section below). Overall, we ensured one-to-one mapping of homologous cell types among the studied species for benchmarking purpose, even though this might lose granularity and lead to slightly more coarse annotation in the heart tasks.

### The BENGAL pipeline

The BENGAL pipeline was built using Nextflow (v22.04.3) DSL2 in java (OpenJDK v11.0.9.1-internal) and utilises singularity containers[44] for portability and reproducibility and is compatible with the execution on high performance computing (HPC) clusters. For data integration, python-based scripts were written based on Python (v3.9.13), Scanpy (v1.9.1), h5py (v3.7.0) and anndata (v0.7.5). We used python implementation of harmonypy (v0.0.5), scanorama (v1.7.2), scVI and scANVI (v0.15.0 with pytorch v1.12.1 and cudatoolkit v11.6 to support execution with Nvidia GPU), SAMap (v1.0.2). R-based scripts were based on R (v4.0.5). We used Seurat (v4.1.1), LIGER (v0.5.0) and LIGER UINMF (v1.1.0) and fastMNN (v1.12.3 from package batchelor). For SCCAF analysis, we made modifications to the SCCAF package v0.0.10 and provided a docker container in docker://yysong123/intgpy:sccaf. For batch correction metrics and biology conservation metrics calculation, we used scIB (v1.1.3).

### Translating features cross-species by gene homology

We used ENSEMBL multiple species comparison tool (version 106), accessed via biomaRt (v2.46.3), to annotate gene homology and translate ENSEMBL gene id cross-species. We chose ENSEMBL to be the primary database for gene homology annotation, since there is manually curated homology in addition to the automated annotation pipeline, it is a popular choice among previous and current cross-species scRNA-seq studies and it is broadly accessible for biologists from various disciplines. BENGAL can be conveniently adapted to use other homology databases, such as EggNOG or BLAST, as long as a homology matching table can be provided.

### Construct concatenated cross-species count matrix

Starting from raw count matrices of scRNA-seq data from different species, the BENGAL pipeline first identifies gene groups composed of one-to-one, one-to-many or many-to-many orthologs and species-specific genes using ENSEMBL multiple species comparison tool. One-to-one orthologs are directly matched across species. On top of that, one-to-many or many-to-many orthologs within each gene group are added and matched by selecting those with higher average expression level calculated using scanpy.pp.calculate_qc_metrics, or with stronger homology confidence defined in ENSEMBL using available attributes among: orthology.confidence, Gene.order.conservation.score, Whole.genome.alignment.coverage, query.gene.identical.to.target, gene.identical.to.query.gene in respective order from biomaRt. The method LIGER UINMF also takes unshared features (species-specific genes) and these genes were added in addition to the three types of inputs with shared features. SAMap quantifies gene homology strength using de novo BLAST, so the raw count matrix and the full set of genes were given to the method without the above homology-matching step, and tblastx (from BLAST v2.12.0) was run using transcriptomes data to generate blast maps for the method.

### Standard processing of scRNA-seq data

Per-species data and unintegrated data were analysed following the scanpy standard analysis workflow with following parameters: sc.pp.highly_variable_genes(adata, min_mean = 0.0125, max_mean = 3, min_disp = 0.5); sc.pp.scale(adata, max_value = 10); sc.tl.pca(adata, svd_solver = 'arpack'); sc.pp.neighbours(adata, n_neighbors = 15, n_pcs = 40); sc.tl.umap(adata, min_dist = 0.3, spread = 1).

For the homology concatenated data input to integration algorithms, we selected highly variable genes per integration key by sc.pp.highly_variable_genes(adata, min_mean = 0.0125, max_mean = 3, min_disp = 0.5, batch_key = integration_key). This function calculates HVGs per batch and uses the intersection of HVGs across batches as input for PCA (see below for integration key per task).

### Cross-species data integration

Taking the concatenated matrix with cells from all species and features matched by homology as input, the pipeline runs 10 integration methods using default parameters.

fastMNN: we used the fastMNN from github.com/LTLA/batchelor, via SeuratWrappers (v0.3.0) following the tutorial http://htmlpreview.github.io/?https://github.com/satijalab/seurat-wrappers/blob/master/docs/fast_mnn.html.

harmony: we used scanpy.external.pp.harmony_integrate to run harmony following tutorial: https://scanpy.readthedocs.io/en/stable/generated/scanpy.external.pp.harmony_integrate.html.

LIGER: we used the LIGER method via SeuratWrappers (v0.3.0) following the tutorial: http://htmlpreview.github.io/?https://github.com/satijalab/seurat-wrappers/blob/master/docs/liger.html.

LIGER UINMF: we ran LIGER UINMF via github.com/welch-lab/liger following the tutorial http://htmlpreview.github.io/?https://github.com/welch-lab/liger/blob/master/vignettes/SNAREseq_walkthrough.html.

SAMap: we ran SAMap using github.com/atarashansky/SAMap following the tutorial and vignette in the github repository.

Scanorama: we used scanpy.external.pp.harmony_integrate to run scanorama following tutorial: https://scanpy.readthedocs.io/en/stable/generated/scanpy.external.pp.scanorama_integrate.html.

scANVI: we ran scANVI following documentation https://docs.scvi-tools.org/en/stable/api/reference/scvi.model.SCANVI.html. Note that although scANVI can be initialised from a pre-trained scVI model, we built scANVI models from scratch using the input AnnData object. This is to make the scANVI runs independent of scVI runs for benchmarking purposes.

scVI: we ran scVI from scvi-tools.org/ following tutorial https://docs.scvi-tools.org/en/stable/tutorials/notebooks/harmonization.html.

SeuratV4CCA and SeuratV4RPCA: we ran SeuratV4CCA from https://satijalab.org/seurat/index.html following https://satijalab.org/seurat/articles/integration_introduction.html and RPCA following https://satijalab.org/seurat/articles/integration_rpca.html.

### Integration key

SAMap is designed to perform integration between data from different species so 'species' were used as the integration key. For the integration methods other than SAMap, we used 'species' as the integration key if there is only one batch per species (Embryo_dr_xt task) or per-species batches have unbalanced cell type composition (Pancreas_hs_mm task and Heart tasks), otherwise, we used 'batch' as integration key if each batch has balanced cell types (Hippocampus_hs_mu_ss task). Highly variable genes were selected using the same integration keys.

### Dimension reduction and visualisation

For methods which output an embedding, neighbourhood graphs were calculated with n_neighbors = 15, n_pcs = 40, and the UMAP representations were computed using min_dist = 0.3 and spread = 1.0. Points were shuffled when plotting UMAP to avoid colour overlap. For methods that output a pseudo-count matrix, we first calculated PCA, then used the same parameters to calculate neighbourhood graph and UMAP. SAMap outputs a corrected neighbourhood graph, so we only recalculated UMAP using the same parameter for visualisation.

## Species mixing score and batch correction metrics

Species mixing was assessed by 4 established batch correction metrics for scRNA-seq. This includes the principal component regression (PCR), batch average silhouette width (bASW), graph connectivity (GC) and k-nearest neighbour batch effect test (kBET). Principals of these metrics are described in Supplementary Methods. In practice, we used the scIB package to calculate the metrics[17]. All metrics were computed on the PCA embedding for methods returning a pseudo-count matrix (SeuratV4 methods, n_pcs = 20), or the output embedding for methods returning a latent embedding (n_dims = 20). For metrics that operate on kNN graphs, we calculated $k = 20$ nearest neighbours on the PCA or embedding both using the first 20 dimensions. In addition, PCR does not rely on cell type annotation whilst the other metrics take cell type labels into account. We highlighted the separation of these two types of methods in detailed scores and rankings of each task (Supplementary Figs. 1–7).

We computed these 4 metrics in 28 integration strategies and 3 unintegrated data of different homology strategies. For each metric, we performed min-max scaling to reflect the relative performance of different strategies in concordance with scIB[17]. We included the unintegrated data in the scaling as a reference, to avoid minor differences in metrics between algorithms that performed similarly well enlarged by scaling.

Species mixing score is the average of the scaled 4 metrics. This is to balance the potential biases between different metrics as none of them are on itself a comprehensive evaluation of species mixing.

$$\text{Species mixing score} = \text{average}(PCR_{scaled}, bASW_{scaled}, GC_{scaled}, kBET_{scaled}) \quad (1)$$

Unlike the study of scIB, we did not include batch graph Local Inverse Simpson's Index (iLISI) score. This was because most integration outputs simply have a raw iLISI score near 0 (see Supplementary Fig. 43 for raw metrics distribution). We reasoned that integrated data could not show improvement in terms of iLISI score because cross-species difference is among the largest effects. All other batch correction metrics could show informative range of variation and using the average was the most appropriate for comparing between strategies. See Supplementary Methods for principles of the batch metrics, with further details available in the scIB study[17].

## Biology conservation score and metrics

Biology conservation was assessed by 5 biology conservation metrics for scRNA-seq, as well as ALCS, a novel metric we developed to reflect maintenance of cell type distinguishability. Biology conservation metrics include cell type ASW (cASW), normalised mutual information (NMI), ARI, iso_F1 score and trajectory conservation score (Traj, only applicable in the Embryo_dr_xt task). Principals of these metrics were described in Supplementary Methods. We computed these 5 metrics in 28 integration strategies and 3 unintegrated data of different homology strategies using scIB and perform min-max scaling. Biology conservation score is the average of the scaled metrics.

$$\text{Biology conservation score} = \\ \text{average}\left(cASW_{scaled}, NMI_{scaled}, ARI_{scaled}, isoF1_{scaled}, Traj_{scaled}\right) \quad (2)$$

We did not include cell type graph LISI (cLISI) in scIB for a similar reason with iLISI: this score was easily fulfilled by most of the results in our study (Supplementary Fig. 43). All strategies achieved a cLISI close to 1 and thus was not informative to scale these metrics or include it in biology conservation score. See Supplementary Methods for

principles of the biological conservation metrics, with further details available in the scIB study[17].

## Integrated score

Integrated score was the weighted average of species mixing score and biology conservation score. In concordance with the scIB study, we gave 0.4 weight to species mixing and 0.6 weight to biology conservation.

$$\text{Integrated Score} = 0.4 * \text{species mixing score} + 0.6 * \text{biology conservation score} \quad (3)$$

In this study, we presented integrated scores that allow for comparison between integration strategies utilised in our cross-species integration tasks. It is important to note that scores in this study do not suggest a complete evaluation of the algorithms employed. These algorithms primarily perform cross-batch and cross-modality integration, as demonstrated by their benchmarking on these tasks in the scIB study[17].

## Alignment score

To quantitatively analyse the degree of cross-species alignment achieved by SAMap, we calculated the alignment score (AS) used in the SAMap study. AS is the average percentage of cross-species neighbours over the maximum number of possible neighbours across all cells from all species.

$$\text{Alignment score} = \frac{\sum_{cells} \frac{\text{number of cross−species neighbours}}{\text{maximum number of neighbours}}}{\text{total number of cells}} \quad (4)$$

This reflects the degree of cross-species alignment and is comparable between SAMap and other strategies. We calculated AS on the kNN graph calculated from embeddings or PCA of pseudo-count matrices from other strategies with sc.pp.neighbours(adata, n_neighbours = 20, n_pcs = 20) and directly on the graph output of SAMap (SAMap ran with n_neighbours = 20). Across all tasks, SAMap showed significantly higher AS than other strategies, while other strategies showed smaller improvement over unintegrated data (Supplementary Fig. 17). Undeniably, SAMap directly operates at kNN level to align cross-species neighbours, resulting in enhanced cross-species alignment. On the other hand, we treated inter-species and intra-species edges equally when kNN was calculated for other methods. Nevertheless, AS is in concordance with the observed strong cross-species alignment by SAMap from the UMAP visualisations.

## Assessing biology conservation with ALCS

We propose a new metric, ALCS, to assess the maintenance of cell type distinction after cross-species integration. This is motivated by our observation that integration algorithms tend to overcorrect and lose the separation of similar cell types on the integrated embedding, since cross-species difference is huge. This behaviour contradicts the goal of cross-species integration, as any species-specific population with subtle differences from others will become unidentifiable after integration.

To calculate ALCS, we adapted functions from the Single Cell Clustering Assessment Framework (SCCAF)[32]. SCCAF is a self-projection-based approach to assess the validity of a classification system, in our case the cell type annotations. Self-projection is the process in which a machine learning classifier is trained on half of the training set and used to predict the label of the other half. The accuracy of self-projection indicates the clarity of the classification. In our case, we chose to use multinomial logistic classifiers as it has been shown to perform equally well compared with other models in the SCCAF study

(see Supplementary Note 3 for other supported classifiers and discussions). We explored the possibility of using a kNN classifier for ALCS but concluded that several caveats hinder its proper adaptation so far (see Supplementary Note 4 for analysis). We computed the loss of self-projection accuracy on the integrated embedding compared with the original per-species data, as a measurement of how much cell type distinguishability is lost due to integration.

$$\text{ALCS} = \text{test accuracy(original)} - \text{test accuracy(integrated)} \quad (5)$$

SCCAF models were trained on the PCA embedding for methods returning a pseudo-count matrix and the output embedding for those who output an embedding. It is not applicable for methods that output a kNN graph.

### Aligning ontology annotation with scOntoMatch
ScRNA-seq datasets from different studies often have varying levels of granularity in their cell type annotations, making it difficult to compare them. To address this, we developed an R package called scOntoMatch, which aligns the granularity of cell type annotation. Such alignment requires an ontology as a reference and we used Cell Ontology in this study[37]. CL describes the ancestor-descendant relationships between cell type terms across animals, making it a reliable reference for harmonising annotation granularity for cross-species annotation.

The scOntoMatch algorithm mainly has two steps. First, it trims the cell type tree in each dataset to remove redundant terms. Second, it identifies an ontology matching across the datasets. The latter is achieved by finding terms that can be directly matched and matching descendants in one dataset to ancestors in another to find the last common ancestor (LCA) term across all datasets. The output is an ontology annotation of all cells that has the highest possible granularity, and aligned across datasets, enabling cross-species comparison. In the aligned annotations, every cell type term is a leaf node in the directed acyclic graph (DAG) created by all cell types from all the input datasets. The scOntoMatch package also provides functions for visualising the CL hierarchy. This approach is essential for batch correction and biology conservation metrics, which rely on cell type annotation and require a systematic, structured way of matching cell type annotations across studies.

### Annotation transfer cross-species
To transfer cell type annotations between species, we employed the SCCAF multinomial logistic classifiers. For each pair of species, we initially trained an SCCAF classifier using data from one species on the integrated data. Subsequently, we used this classifier to infer the cell type annotation of the other species within the same integrated result. To evaluate the accuracy of the transferred annotations, we calculated the ARI between the original annotation and the transferred annotation for each species. The overall ARI for each integration strategy was computed by averaging the ARIs of all species in all transfer runs. A successful annotation transfer is indicated by an ARI close to 1.

### Reporting summary
Further information on research design is available in the Nature Portfolio Reporting Summary linked to this article.

## Data availability
All datasets analysed in this study are publicly available. Raw count matrices and published annotations can be download from the following sources: inDrop data from human and mouse pancreas are available via the GEO database under accession code GSE84133[45]; snRNA-seq data from human, macaque and pig hippocampal and entorhinal regions are available via the GEO database under accession code GSE186538[33]; scRNA-seq data of heart and aorta tissue from human are available via figshare [https://figshare.com/projects/Tabula_Sapiens/100973][1], snRNA-seq data of the heart of long-tail macaque are accessible via the NHPCA database [https://db.cngb.org/nhpca/download][3], scRNA-seq data of mouse heart are available via the EBI ArrayExpress database under accession code E-MTAB-8810 (only no compound treatment mouse data was used)[36], microwell-seq data of *Xenopus laevis* heart are available via figshare [https://figshare.com/articles/dataset/Cell_Atlas_of_the_Xenopus_Laevis_at_Single-Cell_Resolution/19152839][46] and microwell-seq data of zebrafish heart are available via the ZCL database [https://bis.zju.edu.cn/ZCL/][5]; inDrops data of zebrafish embryo are available via the GEO database under accession code GSE112294[34] and inDrops data of xenopus embryo are available via the GEO database under accession code GSE113074[35]. Reference transcriptomes used are listed in Methods. Raw metrics, scaled metrics, scores and rankings for all tasks generated in this benchmark are provided in Supplementary Data 1.

## Code availability
The BENGAL pipeline is available at https://github.com/Functional-Genomics/BENGAL and the version of code used in this study is available via Zenodo with https://doi.org/10.5281/zenodo.8268784[47]. Codes and source data for generating the figures in this study are deposited at https://github.com/Functional-Genomics/BENGAL_reproducibility. The package scOntoMatch is available through CRAN https://cran.r-project.org/web/packages/scOntoMatch/index.html, the version used in this study is 0.1.0 and the development version is available at https://github.com/Functional-Genomics/scOntoMatch.

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

## Acknowledgements
This work was supported by: the European Molecular Biology Laboratory (Y.S., Z.M., A.B., I.P.); the EMBL international PhD program (Y.S.); and the Biotechnology and Biological Sciences Research Council (BBSRC) grant 'Fly Cell Atlas' [BB/T014563/1] (I.P.).

## Author contributions
All authors conceived the project. Y.S. designed the framework and curated data, wrote the pipeline and software, performed data analysis and visualisation, with supervision from I.P. and inputs from Z.M. and A.B. Y.S. wrote the initial draft of the manuscript and all authors participated in review and editing.

## Funding

## Competing interests
The authors declare no competing interests.
