## [Peer Review File · Nature Communications]

Benchmarking strategies for cross-species integration of single-cell RNA sequencing dataREVIEWER COMMENTS

Reviewer #1 (Remarks to the Author):

The authors present a benchmark of cross-species integration of scRNA-seq data considering 9 integration methods combined with 4 between species gene mapping approaches. This is a difficult problem due to the strong species effect, presence of species specific populations and the difficulty of mapping genes and cell types between species. To implement the benchmark the authors developed a Nextflow pipeline called BENGAL (Benchmarking ENsemble Gene mapping And Integration of scRNA-seq data Across species and species pairs). The approaches are evaluated using previously established metrics for assessing batch correction and the conservation of biological variation. The authors also propose a new metric for assessing the conservation of biological variation, the Accuracy Loss of Cell type Self-projection (ALCS) which measures the ability to distinguish cell types within a species after cross-species integration. They also developed a method called scOntoMatch which can be used to harmonize cell ontology labels across species or datasets. The results of the benchmark suggest guidelines for which methods to use in different scenarios, specifically scVI and Harmony for general cases, Seurat for when species are distantly related and SAMap for complex cases such as whole body atlases. Overall I think the benchmark is well designed and the authors do a good job of presenting the limitations and an outlook for further comparisons and method development.

Major concerns:

- The process for scaling, aggregating and ranking metrics is not explained in the methods section. You should clearly explain how this was done for individual metrics, metric categories (batch correction, biological variation, etc.), the overall score and ranking across datasets. Some of this information is in a figure caption but how this is done can have a big impact on the results and should be covered in detail in the methods.
- This description should also cover the weighting of metrics in the overall score, particularly why you chose to weight the ALCS score so highly. I find it odd that you would assign 75% of the biological variation score to a single metric.
- This is a comment that generally applies to these kinds of benchmarks but was not covered in the limitations. With the kind of scaling you have used you can really only see relative performance of methods, not absolute performance, and it may be the most methods perform similarly well (or poorly) but minor differences are emphasised by the scaling. You may want to consider adding artificial good/bad performing methods to help assess possible range of performance.
- In general there were some claims made that were not linked to evidence (or at least the link was not clear). For example, it was not clear what the evidence for the claims in L110-115 was. Please make sure

that all claims (including those not mentioned in this review) are supported by the data and that the evidence is clearly presented.

Minor concerns:

- Most of the discussion in the paper focuses on the integration methods, could you expand a bit on the differences between the ortholog mapping approaches? I think in the long run this may have as large an impact.
- Have you considered evaluating a method that incorporates cell type annotation in the integration step such as scANVI? It's not necessary for the paper but I think it would be interesting to see if that helps overcome some of the issues you see such as the difficulty of integrating between distant species.
- When you select highly variable genes is that done across all the data being integrated or separately for each species? This may have an effect if different species use genes in different ways for example.
- Which batch key do you use for metrics? Is it the key used for integration or always species?
- Have you checked the quality of integration within a species? This is secondary to the integration between species, but it would be interesting to see if methods can still perform well at this easier task while performing good species integration.
- I think the explanation of the ALCS score could be improved, it's not quite clear to me how it is calculated. Some formulas may help here.
- How is ALCS run on methods that output a graph? I wasn't sure how that would work from the description.
- The metrics for assessing label projection are not described, this should have a methods section.
- In Figure 2 (and similar tables in other figures) the use of different symbol shapes for the mapping approaches makes it difficult to compare methods. I would suggest using one symbol and adding the output type as another column (or leaving it off as it is already included in the row labels).
- I found the term "Integrated score" a bit confusing (at first, I thought it meant the average of integration metrics alone). I would suggest using something like "Overall score" instead.
- When looking at the UMAPs in Figure 3 it is unclear why SAMmap was a poor performer based on batch correction metrics (the integration between species looks better than the others). I know UMAPs are highly summarised so maybe you can explain this from the metric scores?
- I think the UMAPs for SAMap in Figure 3F have been switched (cell types and species are the wrong way around)
- Figure 5 A and B show some very similar UMAPS. Could this be visualised in another way or at least the differences between them highlighted somehow?

- I found the diagrams in Figure 5C very difficult to follow, it is difficult to see where the arrows are pointing to and the widths are indistinguishable. I think that something simpler like a heatmap would be much clearer.
- I found the detail of the scOntoMatch algorithm lacking and difficult to understand. It would be good to expand the description of this to make it clearer.
- The plots in Figure 4C don't highlight any biology and it's not clear what cell types etc. are being referred to in L157
- Figure 5 only shows results from Seurat but the text describing that section mentions other methods for which there are no plots shown
- In the discussion you mention that RPCA is more scalable than CCA for the Seurat method but no scalability data is shown. If that comes from the Seurat documentation it should be clearly mentioned.
- I think you use the terms ALCS and "accuracy loss" interchangeably. It might be clearer to only use one or state in the text that they are the same thing.
- In L13 you talk about "batch effects" and the "species effect", it would be good to clarify which of these you consider to be technical/biological.
- Citations for LIGER are missing on L20
- In the discussion you suggest things that future methods could consider (homology matching between genes etc.). My understanding is that SAMap already considers many of these things. It would be good to clarify if this is the case or which things apply to SAMap.
- Figure 1 implies that you perform quality control as part of the pipeline, but the methods and Table 1 state that you use quality control from the original authors. It would be good to make sure this is clear and consistent in the methods/figure.
- Table 1 should include the number of samples per dataset. The specific technology information is also missing from some of the entries (given only as scRNA-seq).
- It is surprising how few many-to-many mapping are mentioned in Table 1. Perhaps it can be clarified what you were listing here as from my experience mapping between species I would expect 100-1000s of these.

Typos and grammar:

- Please double check that the spelling and capitalisation of method names are consistent (and preferably uses the same format as the method authors)
- Ranges for values should be low to high (0.9-0.95), there are quite a few times in the text where they are high to low (0.95-0.9)
- The sentence on line L189 is unclear, I think there might be a missing/extra word

- There were a few sentences in the paper where a word was missing, this didn't affect understandability, but checking for these would help with readability
- In the caption for Figure 1 some of the sentences don't start with a capital letter

Code and software:

- The code and software was available at the referenced repositories
- It would be good to add the Dockerfile to the pipeline repository
- Please add a license to the reproducibility repository
- It would be good to add a bit more content to the reproducibility README explaining what is in the repository and in what order the scripts should be run

Reviewer #2 (Remarks to the Author):

This paper by Song and coworkers presents a benchmarking effort to compare existing computational methods in mapping single-cell transcriptomic datasets across species. This is an area of rapidly growing interest and demand, therefore this work can provide a timely and useful guide for a broad audience. Below are a few technical comments/concerns.

1, my largest concern is that cell types do not necessarily have 1-to-1 mapping across species due to cell type duplication or cell type divergence after speciation. This is the key difference from integrating datasets from the same species. But this seems to be the basic assumption of most metrics used in the paper. Selecting methods based on this assumption of 1-to-1 cell type mapping can cause big misconception of biology in many applications.

2, many of the metrics heavily depend on available cell type annotation. However, some methods here reconstruct manifolds in different ways, which often lead to differences (sometimes significant!) in assigning specific cells with cell type labels. This means that the metrics relying on cell type annotations would have bias towards the method that was used for annotating the dataset initially. I suggest to report separately the metrics independent and dependent of cell type annotations.

3, figure 2 is a bit convoluted, as the mapping results of different methods may depend on evolutionary distances. I would suggest to report each task separately (some condensed version of Extended figure 1-

4), and discuss this important point explicitly. In addition, it is really hard to compare the size of different types of symbols.

4, many applications rely on transcriptomes with no orthology annotation, how do different methods perform without it?

5, figure 3F, although the figure says the methods are shown in decreasing integrated score from left to right, but it appears differently to eyes. In fact, SeuratCCA and SAMap seem to have the best species integration than any other methods (SAMap plots seem to be flipped in order). Same comment for Fig. 3C. Can authors reconcile this inconsistency? It seems that iLISI is particularly low for SAMap for both datasets whereas all other batch metrics are high. What's the explanation for this incompatibility? This leads to my bigger concern how the integration scores are calculated. The weights of different metrics used for calculating these scores seem to be quite arbitrary. And why not showing the embryo task as a main figure as well (current extended figure 10)?

6, Related to my last comment, I am also confused with figure 4E. It is pretty apparent that Harmony and fastMNN did a bad job in integrating across species even for the simplest HM task, why are they shown as top methods? The plots are inconsistent with the text (line 146-155), which suggests that harmony and fastMNN have high integration scores. It seems that the frog data is the major outlier, does it integrate well with any other dataset in pairwise integration? Also it seems that only SAMap can resolve mesothelial cells (orange population) and fat cells (the silver population) based on the UMAP, is it correct?

7, figure 5C, ARI is quite low for fish. Can any other method help? Why was this analysis only done for SeuratCCA? Also strangely frog dataset seems to be least integrated (Fig. 4), why did the transfer of labels work better than fish?

8, it is noted that SAMap doesn't provide final integrated manifolds/embedding, how is PCR metric calculated for SAMap? Similarly, SCCAF also needs a PCA embedding, how was ALCS calculated?

9, line 449, Louvain clustering is used for ARI, but Leiden clustering is the gold standard now. Does clustering method affect ARI?

10, the main code isn't publicly available yet, making it difficult to evaluate what has been done.

Reviewer #3 (Remarks to the Author):

The authors benchmark 25 data integration methods using several cross-species data sets, and provide recommendations on which methods to use for closely related versus further species, as well as how to match genes (one-to-many, many-to-many homologous) among species for data integration.

- Benchmarking metrics and figures are highly arbitrary, incomprehensive and statistically unspecific such as: "Integrated score is $0.4 \times$ species mixing score and $0.6 \times$ biology conservation score. Species mixing score is the average of 4 batch metrics, while biological conservation score is $0.75 \times$ ALCS plus $0.05 \times$ for the other 5 metrics."

- The benchmark is too general and discussed only at a high level. It lacks biological context and implications of using a good/bad data integration method.

- Mostly lacks clues and insights why any method is good/bad at something.

- "Batch score"/"Species mixing score" as well as "Biological conservation score"/"Bio score" interchangeably used, potentially causing confusion.

- In Figure 3, SAMap cell types/species plots are shuffled in the wrong row.

- The correct citation to fastMNN is (Haghverdi et al. 2018) not (Zhang et al. 2019).

Overall, the manuscript provides a good cross species data collection/resource and points out a few factors that should be considered for integration of cross-species data. However, the structure and analysis methods do not present a mature and insightful benchmarking of the existing tools, and this in my opinion is not sufficient for publication in Nat. Com.

Responses to reviewer comments

Benchmarking strategies for cross-species integration of single-cell RNA sequencing data

Yuyao Song ^{1,*}, Zhichao Miao ^{1,1}, Alvis Brazma ¹, Irene Papatheodorou ^{1,*}

¹ European Molecular Biology Laboratory-European Bioinformatics Institute (EMBL-EBI),
Wellcome Genome Campus, Hinxton, United Kingdom

* Corresponding authors

Emails:

Yuyao Song: ysong@ebi.ac.uk

Irene Papatheodorou: irenep@ebi.ac.uk

Address:

European Molecular Biology Laboratory-European Bioinformatics Institute (EMBL-EBI)

Wellcome Genome Campus

Hinxton, Cambridgeshire

CB10 1SD

United Kingdom

Tel:

+44 (0)1223 494 444

¹ Current address: Guangzhou Laboratory, Guangzhou International Bio Island, Guangzhou 510005, China.

Table of Contents

Overview by the author:	2
Responses to reviewer #1:	3
Responses to reviewer #2:	19
Responses to reviewer #3:	27
References	30

Overview by the author:

We would like to express our sincerest gratitude for the comments from all reviewers, which led to significant improvements in this manuscript. Based on the reviewer's suggestions and our goals to make this study more contributive, we made the following revisions, to improve the analysis, presentation and discussion:

For the analysis performed:

1. We thoroughly revised the scaling of metrics and the calculation of benchmarking scores. This is related to points 1-3, 5-6 and 1 by reviewers 1, 2 and 3, respectively.
2. We reached out to the authors of the benchmarking metrics in the scIB study ¹ (DOI: <https://doi.org/10.1038/s41592-021-01336-8>) to discuss improvements in metrics usage. We reached an agreement on which metrics to use for cross-species analysis based on the metrics' power and applicability. This is explained in response to point 3 by reviewer 1 and point 5 by reviewer 2 in detail.
3. We discussed with the authors of SAMap ² for a fair evaluation of the algorithm's output. Details are addressed in response to point 5 by reviewer 2.
4. We added a semi-supervised integration algorithm, scANVI, upon suggestion by reviewer 1 in point 6. This expanded our analysis to address when cell type annotation is available, whether supplying this information can improve integration among distant species.
5. We explored the impact of divergence time between integrated species on integration outcomes by adding 9 new integration tasks (pairwise integration of heart tasks).
6. We added a new batch correction metric, kBET. This is because updates of this metric enabled its application to cross-species integration tasks when our manuscript was under review.
7. We added a new biological conservation metric, isolated label F1 score, to examine strategies' capacity of handling cell types that appear only in some of the batches.

8. We performed deeper analysis to relate the method's principle to their observed behaviour, especially in the heart tasks which presented the most challenges.
9. We highlighted the analysis of ALCS, as this metric has drawn significant interest and was being used by the community to evaluate over-correction in cross-species integration for developing new algorithms³.
10. We expanded the analysis of annotation transfer results and showed more results in the main and supplementary figures.

For the manuscript:

11. The main figures 2, 3, 5, 6, and 7 had major updates and other main figures were updated in response to reviewer's comments. We have also generated 47 supplementary figures to support all the analysis results.
12. We enriched the Methods section and added formulas to aid interpretation. We have also added further details regarding benchmarking metrics to supplementary methods.
13. We thoroughly improved the precision, consistency and clarity of the manuscript text.
14. We provided all integrated results for 16 tasks by 28 strategies, in the form of Uniform Manifold Approximation and Projection (UMAP) visualisations, in Supplementary Fig. 31-46. All benchmarking scores, metrics and rankings were provided in Supplementary table 1.

In the manuscript, we highlight the texts that were revised in **red**, and texts that were originally in the manuscript but were moved or rewrote in response to the comments in **blue**. Please see below a point-to-point response to the reviewer's comments, with the original comment in **bold text** followed by our response.

Responses to reviewer #1:

The authors present a benchmark of cross-species integration of scRNA-seq data considering 9 integration methods combined with 4 between species gene mapping approaches. This is a difficult problem due to the strong species effect, the presence of species-specific populations and the difficulty of mapping genes and cell types between species. To implement the benchmark the authors developed a Nextflow pipeline called BENGAL (Benchmarking ENsemble Gene mapping And Integration of scRNA-seq data Across species and species pairs). The approaches are evaluated using previously established metrics for assessing batch correction and the

conservation of biological variation. The authors also propose a new metric for assessing the conservation of biological variation, the Accuracy Loss of Cell type Self-projection (ALCS) which measures the ability to distinguish cell types within a species after cross-species integration. They also developed a method called scOntoMatch which can be used to harmonise cell ontology labels across species or datasets. The results of the benchmark suggest guidelines for which methods to use in different scenarios, specifically scVI and Harmony for general cases, Seurat for when species are distantly related and SAMap for complex cases such as whole body atlases. Overall I think the benchmark is well-designed and the authors do a good job of presenting the limitations and an outlook for further comparisons and method development.

We are very grateful for the reviewer's comprehensive summary of the main results of this study, and for recognizing our in-depth discussion on this subject. Moreover, we highly appreciate the reviewer's constructive, detailed suggestions both in terms of analysis and the presentation of the results. The reviewer has also pointed out many details that could cause potential confusion for the readers, and we believe that our study benefited significantly from clarifying these points.

Major concerns:

- 1. The process for scaling, aggregating and ranking metrics is not explained in the methods section. You should clearly explain how this was done for individual metrics, metric categories (batch correction, biological variation, etc.), the overall score and ranking across datasets. Some of this information is in a figure caption but how this is done can have a big impact on the results and should be covered in detail in the methods.**

We would like to thank the reviewer for pointing out that the processes of scaling, aggregation and ranking of metrics were missing in the methods. We added this information in line 85 to 94 in the main text and further details in line 489 to 539 in methods. Formulas were added to aid comprehension.

- 2. This description should also cover the weighting of metrics in the overall score, particularly why you chose to weight the ALCS score so highly. I find it odd that you would assign 75% of the biological variation score to a single metric.**

We agree with the reviewer that a more balanced approach to combining different metrics for biology conservation is necessary. In response to your comment, we have revised the calculation of the biology conservation score. Specifically, we now take the average of the applicable metrics among cASW, NMI, ARI, iso_F1, and Traj, which aligns with the approach used in sclB. We have provided further details regarding aggregation of benchmarking metrics to benchmarking scores in our Methods section from line 500 to 507, 520 to 529 and 530 to 534.

We initially allocated a high weight to ALCS due to its critical importance in cross-species integration settings, as it evaluates the degree of blending between cell types due to integration. In response to the feedback, we report the ALCS metric results in a separate paragraph, between lines 133 and 150. We enriched the analysis of ALCS by presenting strategies' ALCS in reference integration tasks with different maximum divergence time between species, as well as showing specific examples with UMAPs in the new Fig. 3.

- 3. This is a comment that generally applies to these kinds of benchmarks but was not covered in the limitations. With the kind of scaling you have used you can really only see relative performance of methods, not absolute performance, and it may be the most methods perform similarly well (or poorly) but minor differences are emphasised by the scaling. You may want to consider adding artificial good/bad performing methods to help assess possible range of performance.**

We really appreciate the reviewer for pointing out the impact of scaling on the interpretation of strategy performance. In response, we have taken steps to improve the robustness of our analysis. First, we have revised our scaling and aggregation methods and have added these revisions in the Methods section from line 500 to 507, 520 to 529 and 530 to 534. We included a reference for raw metrics scores for scaling, and we revised the metrics used based on their discriminative power in our benchmarking tasks. Moreover, we have explored an alternative approach of metrics aggregation, and have demonstrated that strategies' rankings remain robust to different ways of aggregating benchmarking metrics. To further enhance the reliability of our analysis, we have sought guidance from the authors of sclB, the current most

comprehensive benchmark for scRNA-seq integration. To elaborate, we have obtained the following results:

- 1) We calculated the metrics on three types of homology concatenated, unintegrated data, and added them to the scaling and ranking process. This is to use the unintegrated data as a reference to put the metrics score into context. In general, we observed that some algorithms did not achieve integration, showing lower integrated scores than unintegrated data. Strategies still show a wide range of variation in terms of performance in all tasks, as shown in the new Fig. 2.
- 2) We examined the range of variation of raw metric scores to find out if the value differences between strategies were informative. As our integrations were performed in challenging cross-species settings, we anticipated that some of the metrics might be difficult to fulfil.

We found out that batch graph iLISI metrics were in most cases close to 0, while the cell type graph cLISI metrics were almost always 1 (Fig. 1). This includes unintegrated data, suggesting that integrated data does not show much improvement compared with no integration. If these scores are min-max scaled, they could enlarge minor differences among algorithms. Including scaled graph LISI metrics in the averaging to obtain benchmarking scores and to rank strategies is thus less appropriate.

Figure 1 Aggregated raw benchmarking metrics. Showing the distribution of raw benchmarking metrics across 16 tasks from 27 integration strategies and 3 types of homology concatenated, unintegrated data. ILISI and cLISI were removed in the metrics aggregation and ranking process due to a small range of variation. Batch corrections metrics include PCR; iLISI; GC; bASW; kBET; biology conservation metrics include ARI; NMI, cLISI; cASW; Traj. PCR, principal component regression; iLISI, batch graph Local Inverse Simpson's Index; bASW, batch average silhouette width; GC, graph connectivity; kBET, k-nearest neighbour batch effect test; NMI, normalised mutual information; ARI, Adjusted Rand Index; cASW, cell type ASW; iso_F1, isolated label F1 score; Traj, trajectory conservation score.

- 3) We discussed this issue with the authors of graph iLISI and cLISI¹ and agreed that in practice, the iLISI score is particularly hard to fulfil, and the cLISI score is often easy to fulfil. The authors also concurred that these two metrics should only be included when they are informative to the benchmark and show meaningful variation across the strategies. Hence, we removed iLISI and cLISI metrics from the scaling and ranking because they are not informative in our cross-species tasks.
- 4) To ensure strategies' rankings are robust to the metrics aggregation approach, we also performed another type of metrics aggregation and found that the rankings remained consistent. We calculated the accumulation of all

metrics without min-max scaling, and ranked the strategies based on aggregated scores. We found that overall, relative ranking between 27 strategies did not change, with or without iLISI and cLISI in the aggregation (Spearman's rank correlation coefficient 0.91 ± 0.06 across 16 tasks for aggregation with LISI scores, 0.93 ± 0.05 for without LISI scores). The exact scores and ranks and alternative rank by different aggregation approaches were provided in Supplementary Table 1, and https://github.com/Functional-Genomics/BENGAL_reproducibility.

In summary, we omitted graph LISI metrics and kept using the scale-then-average approach to aggregate the metrics into benchmarking scores. This was supported by authors of sclB¹ and is currently used in practice in benchmarking efforts such as Open Problems in Single Cell Analysis (<https://openproblems.bio/>).

- 4. In general there were some claims made that were not linked to evidence (or at least the link was not clear). For example, it was not clear what the evidence for the claims in L110-115 was. Please make sure that all claims (including those not mentioned in this review) are supported by the data and that the evidence is clearly presented.**

'In both cases, harmony, fastMNN, scVI, SeuratV4 CCA,

111 SeuratV4 RPCA and SAMap were able to integrate the data while scanorama, LIGER and LIGER

112 UNIMF did not integrate some homologous cell type pairs. Interestingly, after data integration, only

113 scVI preserved observable substructures within each cell type cluster (e.g. CA1 SUB, CA2-3, InN) in

114 the hippocampus task (Fig. 3F). SAMap gave a strong cross-species alignment, overlapping almost all

115 shared cell types.

'

We thank the reviewer for the suggestion on providing more support for the analysis results. We revised our manuscript thoroughly and added relevant evidence into 47 supplementary figures, making sure to present at least one figure for key results. Results originally in L110-115 were revised in the manuscript text line 167 to 170.

Minor concerns:

5. Most of the discussion in the paper focuses on the integration methods, could you expand a bit on the differences between the ortholog mapping approaches? I think in the long run this may have as large an impact.

Thank you for the suggestion on expanding the analysis between ortholog mapping approaches. In terms of benchmarking scores, we only observed a significant improvement in adding in-paralogs in the methods that achieved integration in the Embryo_dr_xt task (results line 258 to 268, Supplementary Fig. 29). We thus performed further analysis to address this point and added results to line 234 to 236, 256 to 257, 268 to 276, discussions to line 293 to 296, as well as figures to Supplementary Fig. 27, 28 and 30.

Although we did not observe enough differences in terms of integrated scores, we performed two more analyses to address the effect of homology mapping from other perspectives.

- 1) We performed 9 additional integration tasks to observe the impact of divergence time between species on integration outcome. Since homology mapping is more challenging for distant species, this has an impact on the integration results. In general, we observed that species that have diverged for a longer time have fewer mapped orthologs and have more in-paralogs compared with one-to-one orthologs (Table 1). Although good-performing algorithms were usually able to perform species mixing in all cases, overall, ALCS and annotation transfer are more difficult for distant species (manuscript line 234 to 236, 256 to 257, Supplementary Fig.27-28). This suggests that biological signals of cell type specificities experience a loss due to the challenging homology mapping.
- 2) We counted the overlap of highly variable genes from three types of homology concatenated data. This is because HVG calling is the prerequisite for data integration algorithms and differences in HVG can affect algorithm performance. We showed the Venn diagrams in Supplementary Fig. 30 and added results in line 268 to 276. Overall, we observed that when integrating evolutionarily distant species, three types of homology data share less common HVGs compared with closely related species.

- 6. Have you considered evaluating a method that incorporates cell type annotation in the integration step such as scANVI? It's not necessary for the paper but I think it would be interesting to see if that helps overcome some of the issues you see such as the difficulty of integrating between distant species.**

Thank you for suggesting adding scANVI to the study. Since scANVI requires cell type information as input, which is not always available (especially when cross-species integration is used to perform cell type annotation transfer), we did not include it in the original manuscript. However, we agree with the reviewer that it would be interesting to see if supplying cell type labels improve the integration for distant species. Hence, we have now added this method to the pipeline and the benchmarks. In general, it performs the best across the board. For integration across multiple species spanning a large evolutionary distance, scANVI also performed significantly better than others. Considering that it is the only semi-supervised method in this study, we discussed the potential improvements by taking cell type annotation into account in discussion lines 287 to 288.

- 7. When you select highly variable genes is that done across all the data being integrated or separately for each species? This may have an effect if different species use genes in different ways for example.**

Thank you for pointing out that this was less clear in the original manuscript. This information was in fact available under the methods for pre-processing and how different integration strategies were run. In response to your suggestion, we now state this explicitly in methods line 444 to 447.

We selected HVGs per batch, and used the intersection of HVGs across batches per integration task to perform PCA. This is done by the function `“sc.pp.highly_variable_genes(adata, min_mean=0.0125, max_mean=3, min_disp=0.5, batch_key=integration_key)”` in the scanpy package. We used the same batch key as those used for integration. Since both “species” and “sample” could be the batch key during cross-species integration, we select the batch key based on whether samples were balanced within each species. Balanced samples share most cell types, hence the HVGs per sample can represent the discriminative features between cell types across the dataset. The intersection of HVGs from all

batches would remove technical effects and be the most biologically informative, thus appropriate for integrating the data. Our criteria are in line with current best practices for scRNA-seq data integration and scIB to obtain optimum integration results ¹.

8. Which batch key do you use for metrics? Is it the key used for integration or always species?

The batch key used to obtain metrics is also the one used to select HVGs and run integration algorithms. In the Hippocampus_hs_mf_ss task, where multiple samples within each species shared most cell types, the integration key was "sample." However, in all other tasks, the samples within each species were unbalanced, and therefore, we used "species" as the integration key in those cases. We have explained this in a separate paragraph in the Methods section between line 475 to 481.

9. Have you checked the quality of integration within a species? This is secondary to the integration between species, but it would be interesting to see if methods can still perform well at this easier task while performing good species integration.

We appreciate the reviewer's interest in the performance of methods on within-species integration tasks. While we agree that within-species integration is important, our focus in this study is to benchmark strategies for cross-species integration, for species mixing, biology conservation and annotation transfer. This has never been systematically performed previously.

We did not include an analysis of within-species integration, because this has been extensively benchmarked in the scIB study ¹ and previous benchmarks ⁴. We believe that our analysis provides a unique contribution to the field and can help researchers make informed decisions on choosing the best strategy for their specific biological interests.

10. I think the explanation of the ALCS score could be improved, it's not quite clear to me how it is calculated. Some formulas may help here.

We added a detailed description of ALCS in methods line 555 to 572. Formula was also added to aid comprehension. Since ALCS is based on our previous work, SCCAF ⁵, we kindly ask readers to refer to that paper for further reference.

11. How is ALCS run on methods that output a graph? I wasn't sure how that would work from the description.

We would like to thank the reviewer for bringing up this point. The calculation of ALCS requires raw count or a feature embedding of scRNA-seq data and it does not run on methods output only a graph. Initially, for SAMap, we used “wPCA” in the SAMap outputs as an embedding to calculate ALCS, according to the details described in the SAMap paper. However, upon further confirmation with the author, this is an intermediate but not the final output of the method. Hence, we updated our manuscript and did not calculate ALCS for SAMap. Please refer to this GitHub issue for the discussion: <https://github.com/atarashansky/SAMap/issues/106>

12. The metrics for assessing label projection are not described, this should have a methods section.

We have now added the explanation of the method for the label projection assessment in a separate paragraph in results line 108 to 111 and methods line 594 to 597.

13. In Figure 2 (and similar tables in other figures) the use of different symbol shapes for the mapping approaches makes it difficult to compare methods. I would suggest using one symbol and adding the output type as another column (or leaving it off as it is already included in the row labels).

We appreciate the reviewer's suggestions on improving the figure for the summary of the strategies' performance. We remade Fig. 2, providing an overall ranking of the integrated scores, as well as the species mixing score and biology conservation score for each task. We also remade Supplementary Fig. 1-16 to show individual scores. We no longer mix symbols and sizes, and we believe that the new Fig. 2 has improved information content and is more comprehensible.

14. I found the term "Integrated score" a bit confusing (at first, I thought it meant the average of integration metrics alone). I would suggest using something like "Overall score" instead.

We would like to thank the reviewer for raising this point. While we appreciate your suggestion to use a different term than "Integrated score," we have decided to keep using it to represent the weighted average of species mixing score and biology conservation score. Our rationale is that we avoided using the term "overall", as we only benchmark with cross-species integration tasks. Most algorithms were built to be within-species integration methods or have other utilities such as cross-modality integration. We try not to give the impression that our assessment of methods is a comprehensive evaluation of their full capacities.

To clarify this point, we have included details and formulas in calculating the integration score in our Methods section from line 530 to 539. This will provide more clarity and transparency for readers to better understand the score.

15. When looking at the UMAPs in Figure 3 it is unclear why SAMmap was a poor performer based on batch correction metrics (the integration between species looks better than the others). I know UMAPs are highly summarised so maybe you can explain this from the metric scores?

Thank you for raising this point. We investigated further the issue of SAMap's species mixing scores not matching the visual perception from the UMAP. We found out that the structure of the output graph of SAMap, and the unique process by which the graph is generated, compared with standard k-nearest neighbour (kNN) graph calculation, made it unsuitable for evaluation by batch metrics. Please refer to response to point 5 by reviewer 2 for a detailed explanation. We added a short description of our findings to line 94 to 97 in results and more information in supplementary methods.

16. think the UMAPs for SAMap in Figure 3F have been switched (cell types and species are the wrong way around)

We have made appropriate corrections in the new Fig. 4.

17. Figure 5 A and B show some very similar UMAPS. Could this be visualised in another way or at least the differences between them highlighted somehow?

Thank you for the suggestion. We have addressed this issue by revising the plots to provide a clearer view of our findings. The main conclusion is that annotation transfer is largely successful when data integration is well-achieved, even between evolutionarily distant species. To support this conclusion, we report the correlation between the ARI of annotation transfer and the integrated score in all tasks in the new Fig. 7. We also provide Supplementary Fig. 26 that shows the ARI relative to the species mixing score and biology conservation score. We use Supplementary Fig. 27 to show UMAP examples of successful and unsuccessful annotation transfers. The second conclusion is that annotation transfer is more difficult between evolutionarily distant species. This is demonstrated by a heatmap in Supplementary Fig. 28. We believe that these changes make our results more clear and easier to interpret.

18. I found the diagrams in Figure 5C very difficult to follow, it is difficult to see where the arrows are pointing to and the widths are indistinguishable. I think that something simpler like a heatmap would be much clearer.

To make the visualisation of annotation transfer results more intuitive, we switched to a dot plot to show the correlation between average ARI and the integrated scores in the new Fig. 7. At the same time, we have taken up your suggestion of a heatmap to show an example of the ARI per species pair in the Heart_hs_mf_mm_xl_dr task in Supplementary Fig. 28. We believe that these changes provide direct support for our two conclusions as outlined in the above point.

19. I found the detail of the scOntoMatch algorithm lacking and difficult to understand. It would be good to expand the description of this to make it clearer.

We added more explanations for the scOntoMatch algorithm in methods line 573 to 589. We have also explained the algorithm to colleagues for feedback on clarity and comprehensibility. More details could be found on the package GitHub page and a

full usage example was presented in the package vignette. We believe that the description of the algorithm is now more comprehensible.

20. The plots in Figure 4C don't highlight any biology and it's not clear what cell types etc. are being referred to in L157

We thoroughly revised the analysis of integration results in the heart tasks, resulting in the new results text line 193 to 236 and Fig. 6. We start off by comparing the scores from different algorithms, to show that biological conservation drops significantly with the addition of species data. Next, we focus on the issue of loss of cell type distinguishability by analysing the iso_F1 scores and the ALCS. We discussed that nearest-neighbour-based methods tend to overcorrect, and cell types become in general less distinctive when more species data were involved in the integration. We were making a similar point with the previous L157, but with a much-improved depth and clarity.

We also show the respective species mixing score and biological conservation score for all UMAPs, so that the quantitative measurements were presented alongside the visualisation. We believe that these additions provide a more comprehensive discussion of the heart tasks.

21. Figure 5 only shows results from Seurat but the text describing that section mentions other methods for which there are no plots shown

We have now generated the new Fig. 7, Extended Data Fig. 26 and 28 to show a higher-level summary of results in all strategies and all tasks, in response to your suggestion. With Supplementary Fig. 27, we further show UMAPs to aid comprehension of a good and a bad annotation transfer with two examples each.

22. In the discussion you mention that RPCA is more scalable than CCA for the Seurat method but no scalability data is shown. If that comes from the Seurat documentation it should be clearly mentioned.

We added citations to Seurat documentation and referred to the scalability analysis in the scIB study for this point in discussions line 306.

23. I think you use the terms ALCS and "accuracy loss" interchangeably. It might be clearer to only use one or state in the text that they are the same thing.

We made the usage of ALCS consistent and removed the usage of "accuracy loss" because they mean the same thing.

24. In L13 you talk about "batch effects" and the "species effect", it would be good to clarify which of these you consider to be technical/biological.

We really appreciate your comment and have now clarified the concepts presented in this study. In our view, the global transcriptomic difference between species in scRNA-seq data is to a large extent biological. This is because tissues of different species have undergone independent complex evolution, resulting in different overall gene expression patterns. We call the global transcriptomic difference between species "species effect" to highlight the cross-species part, in order to distinguish it from pure technical batch effects for samples from the same species. However, this effect is often not distinguishable from technical effects, as different species samples could have been processed with different protocols and sequenced in different situations. Technical effects are almost certainly present when we integrate data from different studies, such as in the heart tasks. We added this explanation of our considerations to the manuscript text line 14 to 15.

The instances in which we used the "batch effect" in the paper were when we specifically refer to technical effects for the same species data, or when we explain the batch correction metrics. It is worth noting that we use these metrics only to have numerical measurements for comparing the degree of species mixing between strategies, and we do not use them to indicate technical or biological differences.

25. Citations for LIGER are missing on L20

We have now added the citations in the new line 24.

26. In the discussion you suggest things that future methods could consider (homology matching between genes etc.). My understanding is that SAMap

already considers many of these things. It would be good to clarify if this is the case or which things apply to SAMap.

Thank you for pointing out that SAMap already handles several limitations mentioned in the discussion. We rewrote this paragraph in discussion line 324 to 346 to distinguish between the challenges that SAMap already addresses, and those that it does not yet handle.

27. Figure 1 implies that you perform quality control as part of the pipeline, but the methods and Table 1 state that you use quality control from the original authors. It would be good to make sure this is clear and consistent in the methods/figure.

We appreciate the reviewer for bringing up this point. We would like to clarify that QC is not part of the BENGAL pipeline. In Fig. 1, we tried to use the grey box to separate pre-processing steps from steps in the BENGAL pipeline, indicated by a blue box. We used figure captions to further explain this separation. To make this clearer, we added emphasis that QC shall be performed in an input-specific manner and is not part of BENGAL in results line 70 to 71 and methods line 371 to 375.

28. Table 1 should include the number of samples per dataset. The specific technology information is also missing from some of the entries (given only as scRNA-seq).

We have added the number of samples and changed the number of cells to the number of cells per batch in Table 1. When we use “species” as the integration key, we consider each species as one batch in the integration. Only for the Hippocampus_hs_mf_ss task, we used “sample” as the integration key, so there are multiple batches per species. We have also updated the specific sequencing technology for all datasets involved in Table 1.

29. It is surprising how few many-to-many mapping are mentioned in Table 1. Perhaps it can be clarified what you were listing here as from my experience mapping between species I would expect 100-1000s of these.

Thank you for raising this point. We have now provided further explanations in the table legends. The "Number of homologous genes analysed in this study" refers to genes that were analysed in the integration tasks. In practice, only genes that meet the following criteria were included: 1) were quantified in all datasets involved, 2) can be mapped to an ENSEMBL gene ID, and 3) have homology annotation across all studied species. Hence, the number of genes analysed was less than the total number of genes annotated in ENSEMBL, due to differences in sequencing and quality control across studies. Additionally, when integrating data from more than two species, we could only include genes with one-to-many and many-to-many annotations among all species as required by integration algorithms. This may result in a smaller number of genes in these categories, especially when distantly related species were involved in the integration. We believe this clarification provides a better understanding of our analysis to the readers.

Typos and grammar:

- 30. Please double check that the spelling and capitalisation of method names are consistent (and preferably uses the same format as the method authors)**

We thoroughly checked for the spelling and capitalisation of method names and made sure they are in line with the author's usage throughout the manuscript.

- 31. Ranges for values should be low to high (0.9-0.95), there are quite a few times in the text where they are high to low (0.95-0.9)**

We have switched for all ranges of values to be from low to high, for example in results line 211 to 214.

- 32. The sentence on line L189 is unclear, I think there might be a missing/extra word**

We rewrote the "Impact of one-to-many and many-to-many homologous genes" section and improved on the clarity of the text. The previous L189 is now rewrote into the new line 259 to 260.

33. There were a few sentences in the paper where a word was missing, this didn't affect understandability, but checking for these would help with readability.

We thoroughly improved the language of the paper to make them more readable. For example, we rewrote results line 259 to 268 for addressing the impact of homology method in strategy's output, discussion line 324 to 340, methods line 556 to 572 for ALCS, methods line 574 to 589 for scOntoMatch, etc.

34. In the caption for Figure 1 some of the sentences don't start with a capital letter

Thank you for pointing this out. We corrected the caption of Fig. 1 and checked for all sentences in figure captions to start with capital letters throughout the manuscript.

Code and software:

- 1. The code and software was available at the referenced repositories**
- 2. It would be good to add the Dockerfile to the pipeline repository**
- 3. Please add a license to the reproducibility repository**
- 4. It would be good to add a bit more content to the reproducibility README explaining what is in the repository and in what order the scripts should be run**

Thank you for the suggestions on the code and software in this manuscript. We have made respective improvements in response to all the above comments. We added the docker hub links to the docker files of algorithms in the BENGAL pipeline to the BENGAL repository. We updated the reproducibility repository considering new analysis performed upon paper revision, providing original results data, Jupyter notebooks and scripts to reproduce the figures, and added a MIT licence. We also improved the README file in the reproducibility repository for more explanations on how to run the scripts.

Responses to reviewer #2:

This paper by Song and coworkers presents a benchmarking effort to compare existing computational methods in mapping single-cell transcriptomic datasets

across species. This is an area of rapidly growing interest and demand, therefore this work can provide a timely and useful guide for a broad audience. Below are a few technical comments/concerns.

We sincerely appreciate the reviewer for recognizing our timely contribution to the field of cross-species comparison using scRNA-seq data. Our study has greatly benefited from the reviewer's in-depth comments on the theoretical concepts and analysis. We believe our paper has improved as a result and can provide as comprehensive as possible guidance for cross-species integration of RNA-seq data for future studies.

- 1. my largest concern is that cell types do not necessarily have 1-to-1 mapping across species due to cell type duplication or cell type divergence after speciation. This is the key difference from integrating datasets from the same species. But this seems to be the basic assumption of most metrics used in the paper. Selecting methods based on this assumption of 1-to-1 cell type mapping can cause big misconception of biology in many applications.**

We would like to thank the reviewer for this insightful comment. We agree that cell type homology between species is not necessarily one-to-one in some cases, especially when cell types have undergone complex evolution. However, it is currently beyond the capacity of any publicly available data integration methods to infer complex cell-type homology, thus this cannot be benchmarked. Moreover, most of the currently published studies that use data integration for cross-species comparison focus exclusively on comparing known one-to-one homologous cell types between species. This is because for most cell types, there is a consensus on which cells bear the same identity between species. For example, the following studies performed cross-species integration and presented integrated UMAP visualisation of cell types with one-to-one homologous annotation from multiple species: primary motor cortex cells in human, macaque and mouse in Fig. 1 ⁶; hippocampal cells in human, macaque and pig in Fig. 2 ⁷; multiple organs in human, monkey and mouse in Supplementary Fig. 16-19 ⁸. We foresee that with the sequencing of more tissues in more species, similar analyses would be performed and published in the near future. However, at the moment, there are no guidelines on which strategy to choose for data integration, which could have a great impact on the analysis and the subsequent conclusions.

We reason that for any integration algorithm to be useful for cross-species analysis, it must at least integrate known one-to-one homologous cell types. For example, algorithms should be able to match mouse pancreatic beta cells with human pancreatic beta cells, which in our study was not successful for some strategies (Supplementary Fig. 31). Only when known one-to-one mapping is achieved, further sub-clustering analysis using the integrated data can be performed to find subpopulations shared or unique for each species. Due to the hierarchical nature of cell type classification, this one-to-one relationship can be at a relatively coarse level. Nevertheless, without a coarse-level mapping, no finer cell type hierarchy can possibly be resolved.

We made sure to highlight this assumption of our study in the introduction line 52 to 59, and in the discussion line 319 to 323. We also discuss the evolving understanding of complex cell type homology in discussion line 311 to 319. We highlight the hierarchical structure of cell type representation as a future direction in discussion line 338 to 342. We believe that our study can be a helpful reference on which integration strategy to choose in primary cases and inspire future work on developing tools which could resolve complex cell type homology.

- 2. many of the metrics heavily depend on available cell type annotation. However, some methods here reconstruct manifolds in different ways, which often lead to differences (sometimes significant!) in assigning specific cells with cell type labels. This means that the metrics relying on cell type annotations would have bias towards the method that was used for annotating the dataset initially. I suggest reporting separately the metrics independent and dependent of cell type annotations.**

Thank you for this excellent point and for suggesting reporting metrics dependent and independent of cell type annotation separately. In our benchmark, the batch correction metric “PCR” and the biology conservation metric “Trajectory conservation score” were independent of cell type annotations, while other metrics were dependent on cell type labels. For PCR, we remade Supplementary Fig. 1-16 for all tasks to show individual metric scores, highlighting the distinction between metrics that rely on cell type annotations (PCR) and those that do not (bASW, GC, kBET). Since no single metric can be comprehensive enough to evaluate species mixing strategies’ output, we average the scaled 4 batch correction metrics to even out

potential biases. This is to obtain a relatively balanced score to rank the strategies, as well as to be in line with the current most comprehensive benchmark study. The Trajectory conservation score is only applicable to the Embryo_dr_xt task. Hence, we analysed this metric separately in the results line 186 to 192 and Supplementary Fig. 19.

- 3. figure 2 is a bit convoluted, as the mapping results of different methods may depend on evolutionary distances. I would suggest reporting each task separately (some condensed version of Extended figure 1-4), and discuss this important point explicitly. In addition, it is really hard to compare the size of different types of symbols.**

We really appreciate the reviewer's comment on the original Fig. 2. We remade the new Fig. 2 to report different tasks separately and did not mix size and shape. Supplementary Fig. 1-16 has also been remade.

To deepen the investigation of the impact of evolutionary distance on integration output, we performed 10 pairwise integration tasks in the heart tasks. Results were put in line 234 to 236, 256 to 257 and 258 to 276 in the results section, discussion line 294 to 297, as well as in Supplementary Fig. 27-30. We also highlight the different impact of evolutionary distance on strategies in the introduction line 46 to 47. Please also refer to our response to point 5 by reviewer 1 for further details.

- 4. many applications rely on transcriptomes with no orthology annotation, how do different methods perform without it?**

We thank the reviewer for raising this point. When there is no orthology annotation available, only SAMap would be able to integrate the data, because it performs de-novo gene-gene mapping cross-species. All the other integration algorithms require the input data to have the same features i.e. genes. Hence, in a cross-species setting, genes have to be matched between the datasets, and this would be done by gene homology. We highlighted this unique capacity of SAMap in discussion lines 306 to 308.

5. **figure 3F, although the figure says the methods are shown in decreasing integrated score from left to right, but it appears differently to eyes. In fact, SeuratCCA and SAMap seem to have the best species integration than any other methods (SAMap plots seem to be flipped in order). Same comment for Fig. 3C. Can authors reconcile this inconsistency? It seems that iLISI is particularly low for SAMap for both datasets whereas all other batch metrics are high. What's the explanation for this incompatibility? This leads to my bigger concern how the integration scores are calculated. The weights of different metrics used for calculating these scores seem to be quite arbitrary. And why not showing the embryo task as a main figure as well (current extended figure 10)?**

We are grateful to the reviewer for pointing out the inconsistencies between benchmarking scores and the UMAP visualisation. To investigate this, we contacted the authors of LISI metrics ¹ and discussed the considerations of using these metrics in cross-species settings, and we contacted the authors of SAMap ² for the applicability of batch correction metrics on SAMap graph output. These investigations result in a deeper understanding of the metrics and algorithms, which we amended relevant results in our manuscript and added extensive details to the Methods section. Below I will outline our findings:

1. We observed the raw, unscaled metric values to examine their discriminative power (Fig. 1 in response to reviewer 1 point 3). From this analysis, we identified that Local Inverse Simpson's Index (LISI) derived scores, including batch LISI (iLISI) and cell type LISI (cLISI), are not suitable metrics to include in the scaling and ranking. They have shown a very small range of variation and thus lack discriminative power.
2. SAMap shows strong cross-species mapping on the UMAPs but has a low species mixing score. Even using the scores adapted to graph output type by scIB, we still observe unreasonably low batch correction metrics. Reading the algorithm details, we hypothesise that the graph output by SAMap has local structures that are formed by cells of the same species.

We contacted and confirmed with the authors that this was indeed the case. Within-species neighbours and cross-species neighbours were treated

differently by SAMap. For each cell, up to 20 within-species neighbours were deliberately preserved to conserve the local topology. This is misleading to batch correction metrics since many neighbouring cells would be from the same species, resulting in a lower score. In contrast, when using the standard approach to calculate kNN on outputs by other strategies, within-species and cross-species neighbours were treated equally. Hence, the absolute batch correction metrics would appear higher than SAMap.

The SAMap authors suggested that since SAMap was not designed to be a batch correction algorithm like the other algorithms, it does not consider the compatibility of batch correction metrics in evaluating its output. The main usage of this algorithm is to provide a way to integrate data when no comprehensive homology annotation is available.

Due to these findings, we made the following changes to our study. Our manuscript has changed accordingly.

1. We removed iLISI and cLISI metrics from the scaling and ranking upon support from the authors of these metrics. Please also refer to the response to point 3 by reviewer 1 for our effort in ensuring the scaling and aggregation process is robust.
2. We no longer use batch correction metrics and biology conservation metrics, except trajectory conservation scores, to evaluate and rank SAMap outputs with other strategies. Instead, we show UMAP visualisations of the integration output alongside other algorithms and use alignment scores (line 94 to 97, supplementary methods and Supplementary Fig. 17) to give a quantitative measurement of the strength of cross-species alignment. Overall, SAMap is a unique method that was specifically designed for cross-species integration, especially when well-curated homology annotation is unavailable. This makes it stand out from other algorithms for particular use cases and we address this strength in the discussion line 306 to 308.

We found that these issues were also observed by other researchers and had led to confusion, such as <https://github.com/theislab/scib/issues/329> for iLISI metrics and

<https://github.com/atarashansky/SAMap/issues/106> for SAMap. Thus, we believe our analysis is helpful in clarifying these issues.

- 6. Related to my last comment, I am also confused with figure 4E. It is pretty apparent that Harmony and fastMNN did a bad job in integrating across species even for the simplest HM task, why are they shown as top methods? The plots are inconsistent with the text (line 146-155), which suggests that harmony and fastMNN have high integration scores. It seems that the frog data is the major outlier, does it integrate well with any other dataset in pairwise integration? Also it seems that only SAMap can resolve mesothelial cells (orange population) and fat cells (the silver population) based on the UMAP, is it correct?**

We agree that the “top5 methods” in the original Fig. 4E was unclear. Since most differences were given by integration algorithms, the results from 3 homology methods of the same integration algorithm looked very similar on the UMAPs. To show results from more algorithms, we took the homology method with the highest integration score from the top 5 integration algorithms. We clarified this in the legends of Fig. 4 and Fig. 5. In addition, we also show all UMAPs from 28 strategies, with respective species mixing scores and biological conservation scores in Supplementary Fig. 31-46. The mesothelial cells (orange) and fat cells (now grey blue) can be resolved also by scANVI and SeuratV4 CCA/RPCA (Supplementary Fig. 4-7).

- 7. figure 5C, ARI is quite low for fish. Can any other method help? Why was this analysis only done for seuratCCA? Also strangely frog dataset seems to be least integrated (Fig. 4), why did the transfer of labels work better than fish?**

We would like to thank the review’s suggestions for showing more results of annotation transfer. We have previously performed annotation transfer analysis on all strategies for all tasks, but only showed SeuratV4 CCA results as an example. In response to your comment, we have included more comprehensive results of annotation transfer across all tasks and strategies in Fig. 7 and Supplementary Fig. 26-28.

Our analysis indicates that annotation transfer is more challenging between species that have diverged for a longer time. We found a strong correlation between the biology conservation score and ARI in annotation transfer, suggesting that the preservation of cell type-specific expression is critical for successful transfer.

Supplementary Fig. 28 presents an example of all algorithms' performance on annotation transfer for all species pairs in the Heart_hs_mf_mm_xl_dr task. The heatmaps revealed that annotation transfers from zebrafish to other species were in general less successful than other species pairs, even for the best-performing algorithm, scANVI. We hypothesise that zebrafish share less cell type-specific expression features with other species, compared with xenopus.

8. it is noted that SAMap doesn't provide final integrated manifolds/embedding, how is PCR metric calculated for SAMap? Similarly, SCCAF also needs a PCA embedding, how was ALCS calculated?

We understand this confusion. Previously, we were using the “wPCA” in the SAMap output data “samap.adata.obsm”, which was an embedding type of output to calculate PCR and ALCS. This is because it appeared as if this was the cross-species joint embedding mentioned in the manuscript and the SAMap kNN was calculated on this embedding in the standard way. However, upon further confirmation with the author (<https://github.com/atarashansky/SAMap/issues/106>), we came to the conclusion that “wPCA” embedding is intermediate and not what kNN was calculated directly upon. Hence, we first removed the metrics that do not apply for kNN type of output.

Furthermore, upon discussion with SAMap's author about batch correction metrics applicability, we conclude that SAMap is not suitable for batch correction metrics assessment due to its behaviour of treating interspecies and intraspecies neighbours differently. After all these investigations, we updated our manuscript and did not calculate batch metrics and ALCS for SAMap. Please also refer to point 5 above, manuscript line 94 to 97 and supplementary methods for more details.

9. line 449, Louvain clustering is used for ARI, but Leiden clustering is the gold standard now. Does clustering method affect ARI?

Thank you for pointing out that the clustering algorithm requires an update. During the revision of this manuscript, the sclB package which we used has changed to use Leiden clustering optimization for calculating ARI and NMI. Hence, we recalculated the clustering optimization with the Leiden algorithm, resulting in updated ARI and NMI scores. Nevertheless, the clustering algorithm used here is to find the best clustering resolution that matches best to cell type annotation. Using a different algorithm does not cause significant changes in the method's scores for NMI and ARI, indicated by Spearman's rank correlation coefficient 0.99 for NMI and 0.95 for ARI between using Louvain algorithm and Leiden algorithm.

10. the main code isn't publicly available yet, making it difficult to evaluate what has been done.

We believe that there is a potential confusion. We would like to clarify that the main code repository was publicly available when we submitted the manuscript. In response to your comment, we further had colleagues checked that they could view the code during this revision. Below, we confirm again the repositories for code used in this study, which is available in the Code availability section in manuscript line 598 to 604.

BENGAL: <https://github.com/Functional-Genomics/BENGAL>

BENGAL_reproducibility: https://github.com/Functional-Genomics/BENGAL_reproducibility

scOntoMatch: <https://cran.r-project.org/web/packages/scOntoMatch/index.html>

Responses to reviewer #3:

The authors benchmark 25 data integration methods using several cross-species data sets, and provide recommendations on which methods to use for closely related versus further species, as well as how to match genes (one-to-many, many-to-many homologous) among species for data integration.

We really appreciate the reviewer's comments on improving the manuscript. We thoroughly revised our manuscript in precision and depth in response to the comments, and our study

has benefited greatly from these efforts. Thank you again for your time and attention to our work.

- 1. Benchmarking metrics and figures are highly arbitrary, incomprehensive and statistically unspecific such as: "Integrated score is $0.4 \times$ species mixing score and $0.6 \times$ biology conservation score. Species mixing score is the average of 4 batch metrics, while biological conservation score is $0.75 \times$ ALCS plus $0.05 \times$ for the other 5 metrics."**

We appreciate the reviewer for pointing out that the weight assignment of scores was arbitrary. We thoroughly revised the calculation of the species mixing score and biology conservation score to be unbiased and robust, also to be in line with the current most established benchmark of single-cell data integration performed in scIB¹. The species mixing score is now the average of 4 batch correction metrics, and the biology conservation score is the average of 5 biology conservation metrics. This is addressed in line 85 to 94 in the main text and in line 489 to 539 in methods. We now also discuss ALCS in a separate paragraph in lines 133 to 150 and Fig. 3 to highlight its importance. Formulas were added for the scores to aid comprehension.

- 2. The benchmark is too general and discussed only at a high level. It lacks biological context and implications of using a good/bad data integration method.**

Thank you for the feedback. While we appreciate your comments, we believe that our study is very timely and valuable to the field of cross-species integration. Our study provided a practical benchmark that covers a wide range of biological scenarios, and we explained the challenges presented by different biological contexts of the tasks in results line 154 to 158, 173 to 175, 194 to 200 and Table 1. We also provide a guideline on choosing integration algorithms based on biological interest, in discussion line 301 to 310. This is so far the most informative, comprehensive guideline on selecting integration strategy based on biology interest.

We addressed one of the most significant problems in cross-species integration, the merging of similar cell types, by proposing and using the ALCS metric. Our approach has already been adopted by a recent study (Rosen et al. 2023), indicating its usefulness to the community.

Moreover, we were the first to systematically perform and benchmark annotation transfer by different integration algorithms. This is important because a key use case of cross-species integration is for guiding the annotation of new species data, while a closely related model organism has an atlas with cell types well-annotated. As scRNA-seq analysis is being performed on more under-studied species, we believe our analysis will be a helpful reference in the long run.

We understand that our study may not focus heavily on biological context, but we designed our integration tasks to be based on different biological interests. We considered all possible reasons for cross-species integration, and our study covers a wide range of biological scenarios. Given the publicly available data and tools, we provide the most comprehensive work on this benchmark subject.

3. Mostly lacks clues and insights why any method is good/bad at something.

We appreciate your suggestion to provide a more in-depth analysis of how the algorithm's principles relate to their observed performance. To improve, we have added the following analysis in response to your concern:

- 1) We discussed the observed tendency of overcorrection from nearest-neighbour-based methods, such as SeuratV4 and fastMNN, in results line 220 to 226 and Fig. 6c.
- 2) We added one paragraph in discussions to specifically link the algorithm's principle to their observed behaviour in our study. This is in line 286 to 292.
- 3) We addressed SAMap's major advantage, which is achieving strong cross-species alignment even when no well-curated homology annotations are available in results line 182 to 186 and discussion line 292 to 294.
- 4) We added a semi-supervised method, scANVI, into our analysis, and suggested that by taking cell annotation into account, scANVI achieves better cross-species alignment than scVI.

We believe that the additional analysis will help readers understand the strengths and weaknesses of each integration algorithm and how their underlying principles relate to their observed performance.

4. **"Batch score"/"Species mixing score" as well as "Biological conservation score"/"Bio score" interchangeably used, potentially causing confusion.**

We appreciate the reviewer for raising this point. We thoroughly checked our manuscript and used the "Species mixing score" and "Biology conservation score" consistently. For the metrics, we also checked that we used "Batch correction metrics" and "Biology conservation metrics" throughout the manuscript.

5. **In Figure 3, SAMap cell types/species plots are shuffled in the wrong row.**

We appreciate the reviewer for pointing this out. We have made corrections to the new Fig. 4f.

6. **The correct citation to fastMNN is (Haghverdi et al. 2018) not (Zhang et al. 2019).**

We appreciate this comment and have corrected the citation in line 23.

Overall, the manuscript provides a good cross species data collection/resource and points out a few factors that should be considered for integration of cross-species data. However, the structure and analysis methods do not present a mature and insightful benchmarking of the existing tools, and this in my opinion is not sufficient for publication in Nat. Com.

References

1. Luecken, M. D. *et al.* Benchmarking atlas-level data integration in single-cell genomics. *Nat. Methods* (2021) doi:10.1038/s41592-021-01336-8.
2. Tarashansky, A. J. *et al.* Mapping single-cell atlases throughout Metazoa unravels cell

- type evolution. *Elife* **10**, (2021).
3. Rosen, Y. *et al.* Towards Universal Cell Embeddings: Integrating Single-cell RNA-seq Datasets across Species with SATURN. *bioRxiv* (2023)
doi:10.1101/2023.02.03.526939.
 4. Tran, H. T. N. *et al.* A benchmark of batch-effect correction methods for single-cell RNA sequencing data. *Genome Biol.* **21**, 12 (2020).
 5. Miao, Z. *et al.* Putative cell type discovery from single-cell gene expression data. *Nat. Methods* **17**, 621–628 (2020).
 6. BICCN. A multimodal cell census and atlas of the mammalian primary motor cortex. *Nature* **598**, 86–102 (2021).
 7. Franjic, D. *et al.* Transcriptomic taxonomy and neurogenic trajectories of adult human, macaque, and pig hippocampal and entorhinal cells. *Neuron* **110**, 452-469.e14 (2022).
 8. Han, L. *et al.* Cell transcriptomic atlas of the non-human primate *Macaca fascicularis*. *Nature* **604**, 723–731 (2022).

REVIEWER COMMENTS

Reviewer #1 (Remarks to the Author):

Thank you for the responses to the reviewer comments and your attempts to address the concerns. We believe this has improved the paper but we have some follow up comments:

- To clarify the comment about assessing integration within one species. We were not suggesting to perform a single species integration and benchmark that, as you point out that has already been done in existing studies. Rather, what we meant is to perform a cross-species integration and then assess how well samples for just one of the species mix with each other. For example to perform a mouse-human integration and then see how well mouse sample mix with each other. This may give a better indication of how well batch effects are removed without confounding with the biological differences between species. While this would be an interesting and useful thing to add to the paper it would be secondary to the main focus of integration between species so is not required.
- ARI is typically used to assess the difference between clustering, perhaps another metric would be more appropriate for assessing classification?
- It would be clearer to use a point and errors bars in Figure 2A rather than adding error bars to a bar plot which is not really designed for that
- The labels in Figure 2B obscure the plot and don't make it any easier to identify specific points. It would be clearer to remove them and maybe highlight points in another way if needed.
- Have you considered reversing the ALCS score (or subtracting from 1)? This would help it fit with the convention that higher scores are "better" and maybe help improve some of the visualisations.
- The labels for the different methods are inconsistent between figures. Sometimes they are spelt out in full and other times there are different abbreviations used. It would be clearer if these were consistent across figures.
- We weren't able to find the Docker files in the GitHub repository, please check they are there.
- There were several small grammatical errors and typos throughout the text, you may want to give it a proofread for these before publication.
- Supplementary Figure 17 does not seem to show raw iLISI scores, please check these are included and the figure reference is correct.
- You may want to consider using another classifier as part of the ALCS metric. Using a kNN classifier would let you apply the metric to methods that only return a graph.

Reviewer #2 (Remarks to the Author):

While the revised manuscript is more technical sound, it is unfortunately less conceptually compelling to me. The paper appears to have evolved into a replication of prior benchmarking studies of data integration methods, albeit with a focus on the application of cross-species comparisons. Below, I elaborate my concerns.

1, it is now apparent that the majority of the benchmarking metrics would be only applicable to data integration methods that are designed to stitch datasets from the same species. However, cross-species data integration is an entirely different problem, which would likely require fundamentally different approaches. This revision has made it clear that the used benchmarking metrics lacks broad applicability.

2, in my last round of review, my biggest concern is that this work relies heavily on some fundamental assumptions that may not be valid in cross species comparisons. But this concern is not addressed in the revised manuscript. For example, the benchmarking relies on the assumption of 1-to-1 cell type correspondence, which is certainly not true especially over long evolutionary distances. The authors claimed that no current methods can address this issue, which is incorrect. If two cell types in a species get mixed with a single cell type in another species after integration, this would suggest a 2-to-1 correspondence instance. However, this type of results would be penalized by the benchmarking metrics used in this study, as some metrics prioritize clear separation of annotated cell types post-integration.

Reviewer #3 (Remarks to the Author):

The revised manuscript addresses several raised questions and the improvements are substantial.

Responses to reviewer comments

Benchmarking strategies for cross-species integration of single-cell RNA sequencing data

Yuyao Song ^{1,*}, Zhichao Miao ^{1,a}, Alvis Brazma ¹, Irene Papatheodorou ^{1,*}

1 European Molecular Biology Laboratory-European Bioinformatics Institute (EMBL-EBI),
Wellcome Genome Campus, Hinxton, United Kingdom

* Corresponding authors

Emails:

Yuyao Song: ysong@ebi.ac.uk

Irene Papatheodorou: irenep@ebi.ac.uk

Address:

European Molecular Biology Laboratory-European Bioinformatics Institute (EMBL-EBI)

Wellcome Genome Campus

Hinxton, Cambridgeshire

CB10 1SD

United Kingdom

Tel: +44 (0)1223 494 444

^[a] Current address: Guangzhou Laboratory, Guangzhou International Bio Island, Guangzhou 510005, China.

Overview by the author:

We wholeheartedly thank the reviewers for their recognition of the improvement of our manuscript upon last revision. We sincerely appreciate their suggestions on additional analysis and explanations of the manuscript. We have performed further data analysis and revised the manuscript accordingly.

In the manuscript, we highlight the text that was revised in **red**, and text that was originally in the manuscript but was moved or rewritten for language clarity in **blue**. Please see below a point-to-point response to the reviewer's comments, with the original comment in **bold text** followed by our response.

Table of Contents

Reviewer #1 (Remarks and Responses):	2
Reviewer #2 (Remarks and Responses):	12
Reviewer #3 (Remarks and Responses):	18
References	18

Reviewer #1 (Remarks and Responses):

Thank you for the responses to the reviewer comments and your attempts to address the concerns. We believe this has improved the paper but we have some follow up comments:

We appreciate the reviewer for the helpful comments and interesting ideas. We believe our analysis or discussions demonstrated below is helpful to explore the current applicability of the ideas and to clarify the logic behind some of the analysis choices.

1. To clarify the comment about assessing integration within one species. We were not suggesting to perform a single species integration and benchmark that, as you point out that has already been done in existing studies. Rather, what we meant is to perform a cross-species integration and then assess how well samples for just one of the species mix with each other. For example to perform a mouse-human integration and then see how well mouse sample mix with each other. This may give a better

indication of how well batch effects are removed without confounding with the biological differences between species. While this would be an interesting and useful thing to add to the paper it would be secondary to the main focus of integration between species so is not required.

Thank you for the clarification. We understand that investigating how the same species samples are integrated in the case of cross-species integration can inform the relative degree of technical and biological correction. However, we recognize that the available data have limited power to comprehensively address this issue. To perform such analysis, we would need several datasets that have different samples from one species, have balanced cell types across these samples, also have data from several species. In our case, only the Hippocampus_hs_mu_ss task data suffices these criteria. In general, most current publicly available datasets do not come with multiple samples per species, or do not include several samples that have an appropriate degree of cell type sharing to enable a robust analysis. Therefore, we think that inferring such information using only one dataset in this case is not sufficient. We are interested in keeping up with the current available datasets to improve the cross-species integration benchmarking efforts to further investigate the extent of technical and biological integration in the future.

2. ARI is typically used to assess the difference between clustering, perhaps another metric would be more appropriate for assessing classification?

We appreciate your feedback regarding the metric to compare annotation transfer. After carefully considering your comment, we would like to explain our rationale for using the ARI metric in this particular context.

In the annotation transfer section, our primary objective is to compare the similarity between transferred labels and the original labels. We are not evaluating the performance of the classifier itself, but rather examining the feasibility of annotation transfer based on the embedding generated by different integration strategies. While we understand that metrics such as precision score or F1 score are commonly used to evaluate classifier performance, our aim is not to compare different classifiers but to compare the two annotations.

The ARI provides a more suitable measure for assessing the similarity of the two clustering because it is corrected for chance. If the embedding serves as a good basis for transferring cell type annotations, ARI will be close to 1. ARI will be 0 if the two clustering have random agreement and can take negative values until -1 if the two clustering are completely

different. In fact, in our study, ARI was negative in the Heart_hs_mf_mm_xl_dr task for LIGER and LIGER UINMF. This is informative in the sense that these methods introduced more error than random label assignment. This information cannot be obtained using the Rand Index, F1 score or precision. Therefore, we conclude that using ARI as an indicator of the annotation transfer quality in our cases is more appropriate.

3. It would be clearer to use a point and errors bars in Figure 2A rather than adding error bars to a bar plot which is not really designed for that

Thank you for the suggestions. We have changed Figure 2A to a boxplot. The meaning of bars, boxes and whiskers are explained in the figure captions.

4. The labels in Figure 2B obscure the plot and don't make it any easier to identify specific points. It would be clearer to remove them and maybe highlight points in another way if needed.

We removed the method labels from Figure 2B. The initial idea was to highlight the top 3 strategies for each task, but we agree that removing them makes the points more clearly identifiable. The top 3 strategies are still discernible on the figure so we do not need to highlight them elsewhere.

5. Have you considered reversing the ALCS score (or subtracting from 1)? This would help it fit with the convention that higher scores are "better" and maybe help improve some of the visualisations.

Thank you for the suggestion. We acknowledge that using (1-ALCS) will result in higher scores indicating better integration performance. However, we appreciate the "loss" concept in the definition of ALCS to indicate unwanted cell type merging. We carefully balanced these two options and implemented the following solution: we provide an option to calculate (1-ALCS) if the metric is to be aggregated with other biology conservation metrics, but we keep the original definition of ALCS and report it as is in this paper. Below we elaborate on our reasoning.

We describe the convention of assessment metrics as (1) range from 0 to 1 and (2) higher score indicates better integration. We noticed that many metrics are scaled to fit the convention with the aim to aggregate them in benchmarking efforts such as scIB¹ and Tran,

H.T.N., et al (2020) ², while the original definitions do not fit the convention. However, the original definition is more consistent with the theoretical concept and with the initial calculation. For example, the Principal Component Regression (PCR) value is defined to indicate the percent variance explained by batch via a linear regression. The PCR value will be low if batches are well-mixed and PCR should be compared before and after integration. Hence, PCR is scaled by $\frac{PCR_{before} - PCR_{after}}{PCR_{before}}$ to obtain a final PCR score that fits the convention; The k-nearest neighbour batch effect test (kBET) metric is 1 if batches are poorly integrated so 1-kBET was used as the final kBET score; graph cell type Local Inverse Simpson Index (cLISI) is subtracted from number of batches (B) and scaled by $\frac{B - cLISI}{B - 1}$ to fit in the convention, etc.

The key concept of ALCS is the decrease of cell type distinguishability due to integration. This directly aligns with the degree of overcorrection by integration algorithms: the higher the ALCS, the more overcorrection. ALCS is calculated by comparing the test accuracy of a logistic classifier before and after integration using self-projection ³ i.e. $ALCS = Test\ accuracy_{before} - Test\ accuracy_{after}$. Since ALCS was first proposed in this study, and we would like to highlight its particular usefulness for cross-species integration, we think it is more appropriate to keep reporting the metric as it is to match the original concept.

Additionally, plotting (1-ALCS) did not offer more visual clarity than ALCS in the original Figure 3. Please see example in Figure 1 below:

Figure 1 Plotting (1-ALCS) for all strategies in the 7 reference tasks. This cannot provide more visual clarity than plotting ALCS (manuscript Figure 3). ALCS, accuracy loss of cell type self-projection; O2O, only use one-to-one orthologs; HE, one-to-one orthologs plus one-to-many and many-to-many orthologs matched by higher average expression level; SH, one-to-one orthologs plus one-to-many and many-to-many orthologs matched by stronger homology confidence.

We are working on submitting ALCS to continuous community benchmarking efforts such as Open Problems in Single Cell Analysis. In this case, we are going to provide an option to calculate (1-ALCS) so it can be easily aggregated with other metrics. This is similar to the “scale” parameter in the metrics functions in scIB ¹. In this study, we are not aggregating ALCS with other biology conservation metrics to highlight its usefulness to assess overcorrection in cross-species integration.

In summary, we would like to keep the theoretical definition of ALCS to highlight its alignment of degree of overcorrection. Moreover, it is reported separately in this work and is practically clearer for the illustrations. We are working on integrating this metric into benchmarking efforts, in which we will provide an option to compute (1-ALCS) for aggregating with other metrics.

6. The labels for the different methods are inconsistent between figures. Sometimes they are spelt out in full and other times there are different abbreviations used. It would be clearer if these were consistent across figures.

Thank you for pointing this out. We have changed the labels in the original Figure 3 (spelled in full) to the abbreviations used throughout the manuscript. We have checked again that abbreviations are used consistently.

7. We weren't able to find the Docker files in the GitHub repository, please check they are there.

We are sorry for the confusion. We previously uploaded the container to DockerHub and provided a download guide here <https://github.com/Functional-Genomics/BENGAL/blob/main/containers/README.md>

. This is because we have already built the container for Linux/Mac machines and it can be directly pulled and run to skip rebuilding the container. The original docker file is now provided on GitHub here <https://github.com/Functional-Genomics/BENGAL/blob/main/ALCS.Dockerfile>

8. There were several small grammatical errors and typos throughout the text, you may want to give it a proofread for these before publication.

We have made several changes to the manuscript text to correct for grammatical errors and typos, also rewrote some parts for clarity. Please see the blue highlighted text in the revised manuscript.

9. Supplementary Figure 17 does not seem to show raw iLISI scores, please check these are included and the figure reference is correct.

Thank you for pointing this out. We were referring to supplementary figure 47 for raw metrics distribution. The reference in the text has been corrected.

10. You may want to consider using another classifier as part of the ALCS metric. Using a kNN classifier would let you apply the metric to methods that only return a graph.

We really appreciate your suggestion regarding using a kNN classifier for the ALCS metric. It is a very nice idea, so we carefully considered the possibility and conducted a thorough analysis. Although we did create a working prototype, we identified several concerns that suggest a significant amount of more work is required to properly extend to a kNN classifier for ALCS. Moreover, we think that not using a feature-by-cell representation deviates from the self-projection concept, which is the theoretical basis of the metric. We are interested to work on this further in the future, but we feel it is currently out of the scope for this paper. In the following, we show our analysis and explain the concerns.

Different kNN graph output generated by algorithms

For methods that generate a neighbour graph output, the final result is a connectivity matrix and possibly a distance matrix. For example, SAMap's primary output is an unweighted connectivity matrix while BBKNN generates weighted connectivity matrix and distance matrix. Since there is no embedding for calculating distances, we thought to directly utilise these matrices in a kNN classifier. A weighted connectivity matrix and distance matrix for other algorithms that generate corrected counts or embeddings can be computed via `sc.pp.neighbors(use_embedding=embedding_key, n_neighbors=K)`. We used the same number of neighbours for calculating a final kNN by different strategies (K=15). It is important to point out that only the kNN output by SAMap is unweighted and makes it an outlier in this case.

Applying a kNN Classifier on Distance Matrices

To apply a kNN classifier on the distance matrix output by integration algorithms, we used the `ad.obsp['distances']` matrix as a precomputed distance matrix to feed into `sklearn.neighbors.KNeighborsClassifier`. We continue using the self-projection principle of ALCS, that is to test how confused is the classifier on distinguishing the different cell types. In contrast with the current ALCS that utilises cell features, the evidence for a kNN classifier would be the classes of cell neighbours.

The first step is to split the training data and the test data per cell type. We divided the dataset into two independent sets by splitting the cells of each cell type by fraction x ($x=0.5$) for training and testing purposes. The kNN classifier was trained using the `n_train X n_train`

matrix and subsequently tested on an $n_{\text{test}} \times n_{\text{train}}$ matrix to compute the test accuracy. The top K neighbours ($K=15$) served as the hyperparameter.

During this stage, we encountered the first issue related to the necessity of omitting part of the data due to the train-test split. The reason is that we cannot re-calculate the distances. Ordinarily, the distance between the test data and the training data is calculated from scratch, but we have to use the already-calculated distance matrix, resulting in ignoring some data. Figure 2 below illustrates this issue.

Figure 2 Schematic of the train test split during application of a kNN classifier for ALCS. Due to the dimension constraints of the training and testing dataset, part of the data in the distance matrix is omitted. ALCS, accuracy loss of cell type self-projection.

Training and Testing Phases of the kNN Classifier

In the training phase, the kNN classifier simply memorises the class labels of the n_{train} samples, as it is a non-parametric classifier. During the test phase, the model observes the supplied distances between the test samples and the training samples, selects the top K neighbours for each test sample, and determines the class of the test sample based on its k neighbours.

It is important to note that the prediction process essentially reduces to a weighted majority vote, with smaller distances conferring higher weights. Note that in `ad.obsp['distances']`, non-neighbours have a distance of 0 due to the design of the `anndata` object, so we replaced these distances with a large number ($1e5$) to assign a minimal weight, ensuring that they are not considered among the top k neighbours.

Since SAMap generates an unweighted connectivity matrix, an unweighted majority vote is performed for this particular algorithm. The classifier counts the top one most abundant classes among the neighbours of each test data and appoints this class as the prediction. The second concern that arises in this phase is that the model is non-parametric. There is no parameter learning from features that trains the model to distinguish different classes. This diverges from the principle of ALCS, with which we want to test how much biological information regarding the distinguishing features between cell types are maintained in the integrated data.

We still calculated the reduction of self-projection test accuracy between integrated data and per-species data as the final ALCS score. Figure 3 shows the kNN-based ALCS score in different tasks:

Figure 3 The resulting ALCS score calculated using a prototype kNN classifier for all strategies in 7 reference tasks. ALCS, accuracy loss of cell type self-projection; O2O, only use one-to-one orthologs; HE, one-to-one orthologs plus one-to-many and many-to-many

orthologs matched by higher average expression level; SH, one-to-one orthologs plus one-to-many and many-to-many orthologs matched by stronger homology confidence.

Results in Figure 3 are in line with the previous version of ALCS shown in manuscript Figure 3, as well as the UMAP visualisations. ALCS sees a global increase with increased divergent time between species for all strategies. LIGER UINMF, LIGER and fastMNN generally have higher ALCS than other approaches. It is not a surprise that SAMap have higher ALCS in two of the heart tasks, because the algorithm is less effective on multiple datasets with many unshared cell types, in line with observed on UMAP. However, it is important to keep in mind that SAMap is an outlier in the sense that it generates an unweighted connectivity matrix.

Other classifiers currently supported by ALCS

The current ALCS supports other machine learning models, such as Random Forest or Support Vector Machine, but all of them operate on feature-by-cell matrices. This is because the theoretical basis of ALCS is self-projection in machine learning. The focus lies in using self-projection to evaluate label quality in the dataset and the degree of separation between different labels. Consequently, changes in the relative accuracy of self-projection primarily stem from class assignment discrepancies. Although the specific classifier employed can impact absolute accuracy, its influence is limited when comparing between classifications. We used a logistic classifier because it showed comparable performance with non-linear models in the original SCCAF paper and offers interpretability if the model is trained using genes as features.

Conclusion

Considering the aforementioned concerns, we feel that much more work is needed to properly adopt a kNN classifier for ALCS. Since the original ALCS with a logistic classifier already generated informative results and enables comprehension. We appreciate your valuable feedback, and our explanations provide a clearer understanding of the rationale behind our decision.

The code used to perform this analysis can be accessed via:

https://github.com/Functional-Genomics/BENGAL/blob/main/bin/sccaf_kNN_distance.py

Reviewer #2 (Remarks and Responses):

While the revised manuscript is more technical sound, it is unfortunately less conceptually compelling to me. The paper appears to have evolved into a replication of prior benchmarking studies of data integration methods, albeit with a focus on the application of cross-species comparisons. Below, I elaborate my concerns.

We express our gratitude to the reviewer's insightful comments. To incorporate the reviewer's suggestions, we have rewritten part of the introduction and discussion as elaborated below. We are aware of the limitations of the data and computational approaches available currently to study cross-species cell type mapping. However, cross-species analysis are being increasingly reported in scientific literature (please refer to examples in response to the first point) and there is an important need for assessments and guidelines, using available data and computational tools. As the first benchmarking study of cross-species integration of scRNA-seq data, we mostly based our metrics on established benchmarking efforts as pointed out by the reviewer. To tailor our benchmark to cross-species circumstances, we developed ALCS as a new metric particularly helpful for assessing cross-species integration, as well as performed annotation transfer analysis. We also analysed a wide range of datasets covering different use cases of cross-species integration. Therefore, we believe that this benchmark is specific and informative for cross-species integration tasks currently being implemented by the community.

1. it is now apparent that the majority of the benchmarking metrics would be only applicable to data integration methods that are designed to stitch datasets from the same species. However, cross-species data integration is an entirely different problem, which would likely require fundamentally different approaches. This revision has made it clear that the used benchmarking metrics lacks broad applicability.

We agree with the reviewer that this study revealed some limitations of current benchmarking metrics on assessing cross-species integration. However, we kindly clarify that the integration and assessment approaches analysed in this study are applicable in a wide range of scenarios, especially for integration among vertebrates. We further believe that the actual integration of scRNA-seq data is the most technically mature and informative in the scenarios that the species have not diverged too far. Below, we demonstrate the current available metrics, explain our effort on properly leveraging established metrics and develop new metrics to have a rigorous benchmark, and explain how the metrics are applicable in our benchmarking cases. To incorporate the reviewer's suggestion, a reflection

about the scope of this paper and the current limitations regarding benchmarking metrics are discussed in manuscript line 342 to 348.

For three alternative types of output by integration algorithms, there are several established benchmarking metrics that are applicable. By taking an average of scaled metrics we maximally remove the potential biases from each individual metric. This is summarised in Supplementary table 2 by Luecken, M. et al. ¹, attached below as Figure 4. For example, metrics including Graph Connectivity, kBET and ARI/NMI works on graph types of output. In this benchmark, we did not include methods that generate kNN graph output except SAMap, such as BBKNN, due to low performance in single species integration. We removed the graph LISI scores due to the metric not showing an informative range of variation across the strategies (Supplementary figure 47). Both of these choices are to obtain helpful and rigorous benchmarking results and we have reached consensus with the developers of scRNA-seq benchmarking efforts ¹.

Metric	Graph	Embedding	Feature	RNA
PCR batch		×	×	×
Batch ASW		×	×	×
Graph connectivity	×	×	×	×
Graph iLISI	×	×	×	×
kBET	×	×	×	×
Normalized Mutual Information	×	×	×	×
Average Rand Index	×	×	×	×
Cell type ASW		×	×	×
Graph cLISI	×	×	×	×
Isolated label F1	×	×	×	×
Isolated label ASW		×	×	×
Cell cycle conservation		×	×	×
HVG conservation			×	×
Trajectory conservation	×	×	×	×

Figure 4 Applicability of benchmarking metrics on different types of outputs by scRNA-seq integration algorithms. Adapted from Supplementary table 2 by Luecken, M. et al. (2021) ¹.

SAMap is the current only published tool that was specifically designed for cross-species scRNA-seq data integration. However, after discussions with the method's developer we reached a consensus that the tool was not designed considering the applicability of current benchmarking metrics. This is because the tool is designed for an entirely different purpose - to stitch whole-body scRNA-seq data from species without a homology annotation. To provide a numerical measurement of integration strength in addition to UMAP visualisations, we have also analysed the benchmarking metric utilised in SAMap, which is the alignment score (Supplementary Figure 17). This metric compares the relative number of cross-species neighbours and can be a starting point for future metrics development.

We further acknowledge that the strategies demonstrated in this study have wide applicability to compare species that have not diverged very far, such as among vertebrates. We believe the actual integration of scRNA-seq data cross-species is also the most useful in these cases. Currently, integration methods that were designed for scRNA-seq integration between the same species data were still used extensively to perform cross-species analysis for studying a single organ. Examples include multiple brain regions such as the cortex ^{4,5}, the dorsal root ganglia ⁶, the hippocampus ^{7,8}, the pancreatic isle ⁹, the eye ¹⁰, the ileum ¹¹, etc., and integration across multiple organs in atlasing efforts ¹²⁻¹⁴. Although current cross-species scRNA-seq integration strategies are limited in the sense of studying evolutionarily distant species, it still provides insightful biology in comparing relatively closely related species in a diversity of systems.

As the reviewer pointed out, for evolutionarily very remote species, we require an entirely different approach, such as to calculate a correlation of cell type marker genes ¹⁵. Or more recently, information from protein sequences ¹³. However, this is a different question with what has been investigated in this benchmark.

There have not been a diversity of tools designed specifically for cross-species integration. Therefore, we do not have an entirely specialised benchmarking metrics system designed for these tools, but rather need to adapt current metrics to obtain a numerical measurement of algorithm performance. In this benchmark, we made the first effort to propose and utilise ALCS as a biology conservation metric that is more meaningful for cross-species integration. However, we are aware that much work is needed to further the development of specialised

cross-species integration algorithms and benchmarking metrics to address more specific issues, especially for integration between evolutionary distant species.

2, in my last round of review, my biggest concern is that this work relies heavily on some fundamental assumptions that may not be valid in cross species comparisons. But this concern is not addressed in the revised manuscript. For example, the benchmarking relies on the assumption of 1-to-1 cell type correspondence, which is certainly not true especially over long evolutionary distances. The authors claimed that no current methods can address this issue, which is incorrect. If two cell types in a species get mixed with a single cell type in another species after integration, this would suggest a 2-to-1 correspondence instance. However, this type of results would be penalized by the benchmarking metrics used in this study, as some metrics prioritize clear separation of annotated cell types post-integration.

We thank the reviewer for this insightful comment. We acknowledge that valid non-one-to-one cell type mappings across species do exist, particularly over long evolutionary distances. Although this would ideally be resolved by scRNA-seq integration, we believe this is a challenging issue for the research community to have a theoretical and methodological consensus. Available computational tools, metrics and ground truth datasets are insufficient to enable benchmarking non-one-to-one mappings. Below, we would like to elaborate our reasoning and explain the approaches in this particular benchmark, in which we fit in the current knowledge framework, provide a rigorous benchmark and investigate limitations. To address the constraints of current computational approaches mentioned by the reviewer, we have now added the discussion between line 331 to 337, line 342 to 348 and line 362 to 368.

We reason that an observed one-to-two mapping in the reviewer's example might arise from several technical and biological possibilities that would be difficult to distinguish with current tools and data. Possible technical reasons include a computational artefact or an experimental bias. From the computational point of view, this could arise due to an integration error. Algorithms are prone to overcorrection and mapping unrelated cell types. An example is the observed cell type merging by LIGER in Figure 3. Another possibility, is an annotation granularity inconsistency. Cell types may be annotated at different granularity between species, resulting in seemingly one-to-many mapping. For example, the endothelial cell of artery and endothelial cell of lymphatic vessels from the human heart data correspond to endothelial cells in all other species data in Supplementary Figure 20, upon which we found necessary to unify the annotation granularity among public datasets. Experimentally, there could be a bias in cell type capturing in one species. Several related populations in the

other species thus seem to map to one cell type in the first species. It could also be due to an incomplete understanding of how many cell types exactly are there are in one species. Current species cell atlas projects are still underway and although they will eventually produce enough data points across organs and individuals to help resolve such issues, their datasets are still patchy. Furthermore, populations with a subtle difference in one species might not be separately annotated but can be distinguished in another species.

The possible biological reasons of non-one-to-one mapping on integrated scRNA-seq data includes a transcriptional convergence or an evolutionary homology¹⁶. In other words, the mapping of cell populations on the integrated data suggests a transcriptomic similarity but not necessarily evolutionary relatedness. It is important to note that the scRNA-seq profile is only one phenotype of the cells. Consequently, any computational method that only operates only on scRNA-seq data cannot provide sufficient evidence to address evolutionary relatedness. At best, solely using scRNA-seq data we can generate hypotheses, but more evidence is required from multimodal data to examine evolutionary homology from shared transcriptional regulatory machinery¹⁷ or other phenotypes. An example of cell type mapping on scRNA-seq data possibly due to transcriptional convergence is demonstrated by Woych, J. et al.⁵ in mammalian neocortex and the dorsal ventricular ridge (DVR) of sauropsids.

Therefore, currently, the above-described biological possibilities cannot be disentangled by a standalone, available computational method, even when technical factors are considered.

In this benchmark, we focus on the computational artefacts introduced by scRNA-seq integration strategies. To assess this, we controlled over other possibilities. We unified the cell type annotation to a consistent granularity between datasets, guided by Cell Ontology, to provide a common ground for applying benchmarking metrics. To have confident evolutionary related cell types that should not be cross-matched, we focused on one-to-one homologous cell types in vertebrates supported by literature and have consensus among the community. Our focus on one-to-one homologous cell types aims to identify algorithms that yield more trustworthy cell type mappings without an integration error, in the most basic one-to-one mapping scenario.

Following these lines, if a top-performing method in this benchmark still generates a non-one-to-one mapping for a particular experiment, this could indicate that there is in fact possible biological correspondence. It is then sensible to continue investigating evidence of evolutionary correspondence using experimental data from other modalities. To this end, our

benchmark helps the researchers to rule out some technical possibilities and choose a better approach to investigate biological questions.

We kindly clarify that The ALCS metric does not measure the absolute degree of cell type separation, but addresses the cell type distinguishability loss after integration by comparing with before integration. We are aware that currently defined cell types can have a continuous transcriptomic profile such as in developmental settings. Hence, we only address artificially introduced cell type merging due to integration. Please note that we only consider one-to-one cell populations that are shared between datasets in ALCS. In the Embryo_dr_xt task, there are several neural crest subtypes in zebrafish that we collectively annotated as neural crests to match xenopus. We did not consider mapping between zebrafish secretory epidermal and xenopus cement gland primordium, or zebrafish rare epidermal subtypes with xenopus small secretory cells or goblet cells inferred by SAMap, because these cell types are not shared between the two species and no further evidence suggest this mapping is evolutionary. Overall, we only considered confidently matched one-to-one homologous cell types in this metric to penalise cell type merging introduced by computational artefact.

In order to establish a benchmark of non-one-to-one mappings in the future, one would need consensus non-one-to-one mapping scenarios backed by additional analysis as ground truth. For instance, If we adopt the evolutionary definition of cell types described by Arendt & Musser et al, this would be a demonstration of sharing of transcriptional regulatory machinery involving TFs encoded by terminal selector genes and their targets. We would thus require matched scRNA-seq and scATAC-seq from species analysed by (to be developed) computational approaches that use both data modalities. However, at this stage, the data necessary for such analyses are still being generated. There is also no computational method that can jointly analyse scRNA-seq data and scATAC-seq data to identify cell type mapping due to evolutionary relatedness. Hence, the ground truth is not available (data and the computational methods) to benchmark non-one-to-one evolutionary cell type mapping.

By benchmarking on datasets with one-to-one cell type correspondence, we can gain valuable insights into algorithm performance and inform the development of methods that handle biological many-to-one or many-to-many correspondences more effectively. Overall, we are mindful of the complexities involved and strive to advance the field by iteratively refining our approaches and incorporating additional analysis as the community generates the necessary data and tools.

Reviewer #3 (Remarks and Responses):

The revised manuscript addresses several raised questions and the improvements are substantial.

We sincerely appreciate the reviewer's comments and we are delighted to know that the manuscript has greatly improved.

References

1. Luecken, M. D. *et al.* Benchmarking atlas-level data integration in single-cell genomics. *Nat. Methods* (2021) doi:10.1038/s41592-021-01336-8.
2. Tran, H. T. N. *et al.* A benchmark of batch-effect correction methods for single-cell RNA sequencing data. *Genome Biol.* **21**, 12 (2020).
3. Miao, Z. *et al.* Putative cell type discovery from single-cell gene expression data. *Nat. Methods* **17**, 621–628 (2020).
4. Bakken, T. E. *et al.* Comparative cellular analysis of motor cortex in human, marmoset and mouse. *Nature* **598**, 111–119 (2021).
5. Woych, J. *et al.* Cell-type profiling in salamanders identifies innovations in vertebrate forebrain evolution. *Science* **377**, eabp9186 (2022).
6. Jung, M. *et al.* Cross-species transcriptomic atlas of dorsal root ganglia reveals species-specific programs for sensory function. *Nat. Commun.* **14**, 366 (2023).
7. Tosches, M. A. *et al.* Evolution of pallium, hippocampus, and cortical cell types revealed by single-cell transcriptomics in reptiles. *Science* **360**, 881–888 (2018).
8. Franjic, D. *et al.* Transcriptomic taxonomy and neurogenic trajectories of adult human, macaque, and pig hippocampal and entorhinal cells. *Neuron* (2021) doi:10.1016/j.neuron.2021.10.036.
9. Tritschler, S. *et al.* A transcriptional cross species map of pancreatic islet cells. *Mol Metab* **66**, 101595 (2022).
10. van Zyl, T. *et al.* Cell atlas of aqueous humor outflow pathways in eyes of humans and four model species provides insight into glaucoma pathogenesis. *Proc. Natl. Acad. Sci.*

- U. S. A.* **117**, 10339–10349 (2020).
11. Li, H. *et al.* Cross-species single-cell transcriptomic analysis reveals divergence of cell composition and functions in mammalian ileum epithelium. *Cell Regen* **11**, 19 (2022).
 12. Han, L. *et al.* Cell transcriptomic atlas of the non-human primate *Macaca fascicularis*. *Nature* **604**, 723–731 (2022).
 13. Rosen, Y. *et al.* Towards Universal Cell Embeddings: Integrating Single-cell RNA-seq Datasets across Species with SATURN. *bioRxiv* (2023)
doi:10.1101/2023.02.03.526939.
 14. Wang, F. *et al.* Endothelial cell heterogeneity and microglia regulons revealed by a pig cell landscape at single-cell level. *Nat. Commun.* **13**, 1–18 (2022).
 15. Tanay, A. & Seb e-Pedr os, A. Evolutionary cell type mapping with single-cell genomics. *Trends Genet.* **37**, 919–932 (2021).
 16. Arendt, D. *et al.* The origin and evolution of cell types. *Nat. Rev. Genet.* **17**, 744–757 (2016).
 17. Arendt, D., Bertucci, P. Y., Achim, K. & Musser, J. M. Evolution of neuronal types and families. *Curr. Opin. Neurobiol.* **56**, 144–152 (2019).

REVIEWERS' COMMENTS

Reviewer #1 (Remarks to the Author):

Thank you for your very thorough responses to the reviewer comments. I believe the examples where you have done further analysis are valuable contributions in their own right that you deserve recognition for and would like to see them published in some way (perhaps as supplementary material).

My only small comment is that the labels in Figure 7 showing the regression coefficients etc. overlap the data points. Please adjust this in the final published figure.

Reviewer #2 (Remarks to the Author):

I appreciate the authors' thoughtful responses and revisions to the text, which are very helpful to understand some of the choices and compromises made in this study. I think this version is ready to publish, but it might help the readers if the authors could choose to incorporate some of the discussion in the reply letter into the main text -- up to the authors though.

Responses to reviewer comments

Benchmarking strategies for cross-species integration of single-cell RNA sequencing data

Yuyao Song 1, *, Zhichao Miao 1, 2, Alvis Brazma 1, Irene Papatheodorou 1, *

1 European Molecular Biology Laboratory-European Bioinformatics Institute (EMBL-EBI),
Wellcome Genome Campus, Hinxton, United Kingdom

2 Guangzhou Laboratory, Guangzhou International Bio Island, Guangzhou 510005, China

* Corresponding authors

Emails:

Yuyao Song: ysong@ebi.ac.uk

Irene Papatheodorou: irenep@ebi.ac.uk

Address:

European Molecular Biology Laboratory-European Bioinformatics Institute (EMBL-EBI)

Wellcome Genome Campus

Hinxton, Cambridgeshire

CB10 1SD

United Kingdom

Tel: +44 (0)1223 494 444

Author overview:

We are grateful for the reviewer for their comments throughout the revision and their support for publication of this study. For the final revision, we reformatted the last paragraph of the introduction to comply with editorial requests and included some analysis in the previous responses into supplementary notes. We have also reformatted the data availability and code availability sections and corrected some details throughout the manuscript and related files. In the manuscript, we highlight the text that was revised in red.

Reviewer #1 (Remarks to the Author):

- 1. Thank you for your very thorough responses to the reviewer comments. I believe the examples where you have done further analysis are valuable contributions in their own right that you deserve recognition for and would like to see them published in some way (perhaps as supplementary material).**

We appreciate the reviewer for the helpful comments throughout the revision. To incorporate your suggestions, we added the analysis of adapting a kNN classifier to ALCS as supplementary notes. Meanwhile, due to the transparent peer review policy of the journal, the reviewer comments to the authors and author responses will be published alongside the article. Hence, we believe that the scholarly value of the reviewer's suggestions and our analysis will be demonstrated through this.

- 2. My only small comment is that the labels in Figure 7 showing the regression coefficients etc. overlap the data points. Please adjust this in the final published figure.**

Thank you for pointing this out. We have adjusted the labels in the final figure 7.

Reviewer #2 (Remarks to the Author):

- 1. I appreciate the authors' thoughtful responses and revisions to the text, which are very helpful to understand some of the choices and compromises made in this study. I think this version is ready to publish, but it might help the readers if the authors could choose to incorporate some of the discussion in the reply letter into the main text -- up to the authors though.**

We appreciate the reviewer's comments and thoughts that helped improve this study. We decided to include some discussions about the current possibilities of seemingly non-one-to-one mappings seen from integrated data in supplementary notes, which may help the readers to be more careful to interpret the integration output. We mentioned these discussions in the main text.